# High-order radiomics features based on T2 FLAIR MRI predict multiple glioma immunohistochemical features: A more precise and personalized gliomas management

Jing Li [1,2¤a¤b], Siyun Liu[3], Ying Qin[3], Yan Zhang[1], Ning Wang[1], Huaijun Liu[1]*

1 Department of Radiology, The Second Hospital of Hebei Medical University, Shijiazhuang, Hebei, China, 2 Department of Radiology, Tangshan Women and Children's Hospital, Tangshan, Hebei, China, 3 Life Science, GE Healthcare, Beijing, China

¤a Current address: Department of Radiology, Shijiazhuang, Hebei, China
¤b Current address: Department of Radiology, Tangshan, Hebei, China
* huaijunliu2@126.com

**Data Availability Statement:** All relevant data are within the manuscript and its Supporting Information files.

## Abstract

### Objective

To investigate the performance of high-order radiomics features and models based on T2-weighted fluid-attenuated inversion recovery (T2 FLAIR) in predicting the immunohisto-chemical biomarkers of glioma, in order to execute a non-invasive, more precise and per-sonalized glioma disease management.

### Methods

51 pathologically confirmed gliomas patients committed in our hospital from March 2015 to June 2018 were retrospective analysis, and Ki-67, vimentin, S-100 and CD34 immunohisto-chemical data were collected. The volumes of interest (VOIs) were manually sketched and the radiomics features were extracted. Feature reduction was performed by ANOVA+ Mann-Whiney, spearman correlation analysis, least absolute shrinkage and selection oper-ator (LASSO) and Gradient descent algorithm (GBDT). SMOTE technique was used to solve the data bias between two groups. Comprehensive binary logistic regression models were established. Area under the ROC curves (AUC), sensitivity, specificity and accuracy were used to evaluate the predict performance of models. Models reliability were decided according to the standard net benefit of the decision curves.

### Results

Four clusters of significant features were screened out and four predicting models were con-structed. AUC of Ki-67, S-100, vimentin and CD34 models were 0.713, 0.923, 0.854 and 0.745, respectively. The sensitivities were 0.692, 0.893, 0.875 and 0.556, respectively. The specificities were: 0.667, 0.905, 0.722, and 0.875, with accuracy of 0.660, 0.898, 0.738, and

**Funding:** The GE Healthcare, Beijing provided support in the form of salaries for author Siyun Liu but did not have any additional role in the study design, data collection and analysis, decision to publish, or preparation of the manuscript. The specific roles of author are articulated in the "author contributions" section. There was no additional external funding received for this study.

**Competing interests:** Siyun Liu receives salaries from GE Healthcare, Beijing. There are no patents, products in development or marketed products to declare. There are no additional competing interests to declare. This does not alter our adherence to PLOS ONE policies on sharing data and materials.

0.667, respectively. According to the decision curves, the Ki-67, S-100 and vimentin models had reference values.

## Conclusion

The radiomics features based on T2 FLAIR can potentially predict the Ki-67, S-100, vimentin and CD34 expression. Radiomics model were expected to be a computer-intelligent, non-invasive, accurate and personalized management method for gliomas.

## Introduction

Glioma is the most common neuroepithelial tumor of the cerebral nervous system. Accurate grading of glioma is meaningful for clinics, however, significantly different prognosis exists among individuals who were classified as the same grade [1–3]. It has been fully studied that gliomas with the same or similar histological characteristics may carried different molecular or genetic information [4–8]. The WHO 2016 classification of CNS tumors also introduces both of the phenotype and the genotype into daily diagnosis and makes it possible to lead a more precise diagnosis of the various entities toward a personalized management of brain tumors. Besides the genomic markers mentioned in the WHO 2016, the concept of molecular diagnosis also promotes immunohistochemical assays to involve more molecular biomarkers besides those for traditional gliomas grading, leading to more precise pathological subtype and better prognostic prediction [9,10]. In recent years, great progress was obtained in the molecular pathology of neuro-tumors and a series of molecular markers have been found to be helpful in the clinical differential diagnosis and prognosis predicting of gliomas, among which Ki-67, vimentin, CD34 and S-100 are four vital biological behavior biomarkers [11–15]. Ki-67, a bio-marker for cell proliferation, has been included in WHO of CNS tumor routinely for grading and prognosis prediction [16]. The other three markers of vimentin, S-100 and CD34 have been studied rarely for gliomas as a single marker and frequently utilized as co-staining histo-logical markers for differential diagnosis or prognostic prediction. For example, the combina-tion of GFAP, EMA, S-100 and vimentin was used to assist epithelioid glioblastoma (Ep-GBM) which is one provisional new variant of glioblastoma added to the WHO 2016 classification [9]. Vimentin is often co-stained with GFAP, Ki-67 and P53 for diffuse astrocytoma as its enhance-ment factors for cell motility and invasion [17], high vimentin expression could be taken as a prognostic factor for treatment difficulty or poor survival [12,18]. S-100 is quite useful in the diagnosis of poorly differentiated tumors thus is often involved in most of glioblastomas immu-nohistological diagnosis [10]. CD34 is popular as a vessel marker and is demonstrated to regulate the glioma angiogenesis and could help gliomas grading [19]. CD34 expression is also candidate as a prognostic biomarker in glioblastoma to identify survival and could also be pre-dictive for efficacy of bevacizumab [20]. These diagnosis guidelines or studies indicate that fully consider the molecular pathology may greatly help confirm gliomas sub-type and prognostic prediction. At present, the pathological histology of glioma after surgical resection or biopsy is the golden standard for gliomas grading and immunohistochemical typing. However, it has some inadequacies such as invasiveness, untimely sampling, time-consuming, sampling errors and different histological interpretations [21]. Therefore, it is necessary to find an effective and non-invasive approach to classify different glioma immunohistochemical subtypes.

Abundant information extracted from radiomics features give us chances to establish bridge between radiological image features and tumor-associated molecules [22]. It is

appearing more and more radiomics-based tools for studying gliomas. Radiomics features derived from MRI images have been used to predict grades of gliomas and showed good performance [23,24]. In addition, texture analysis based on MR images has also been applied in molecular or genomic subtyping and their survival outcome relevance for gliomas [25–27]. The previous studies for molecular or gene subtyping utilized single- or multi-modal MRI features extracted from sequences of T1WI, T2WI, T2 FLAIR, Contrast-enhanced T1-weighted images (T1-CE), advanced MRI techniques such as diffusion-weighted imaging (DWI) and arterial spin labeling (ASL). Among these sequences, T2 FLAIR is one of principle imaging sequence for assessment of gliomas and has the best contrast between infiltrating tumor margins and normal brain [28,29]. And the texture analysis based on T2 FLAIR images has been demonstrated to be helpful and accurate in predicting IDH mutation [30,31]. T2 FLAIR features also showed distinct characteristics between IDH wildtype and mutant tumors,1p/19q co-deleted and 1p/19q intact tumors, MGMT methylated and unmethylated tumors respectively, which could useful in molecular classification of patients with grade II/III glioma [32]. In addition, T2 FLAIR is increasingly involved in the texture analysis for assessment of therapeutic response and survival outcomes. It has been reported that 3D shape features could distinguish pseudoprogression from true progression and lower edge contrast of the FLAIR signal is correlated with poor survival after bevacizumab treatment [33,34].

All of these indicate that T2 FLAIR is candidate to capture some features in the non-enhanced state, and the visible tumor phenotypic or genotypic characteristics can be systematically quantified [35]. Therefore, radiomics analysis of pre-operative T2 FLAIR data is principally functional for identification of immunohistochemical biomarkers, which is often crucial for gliomas pathological sub-typing and prognosis predicting. However, the T2 FLAIR-based texture analysis was rarely studied in glioma immunohistochemical typing research. At present, considerable Ki-67 radiomics studies can be detected [36]. While S-100, CD34 and vimentin have not been discussed yet. In this study, we proposed radiomics features and the binary logistic regression model to identify the immunohistochemical typing of Ki-67, S-100, CD34 and vimentin, so as to achieve the image-indication of tumor progression, angiogenesis, proliferation or invasion. We hypothesized that T2 FLAIR-based radiomics methods would facilitate imaging subtypes of gliomas that relate to prognosis and underlying molecular characteristics of the tumor.

## Materials and methods

### Population

This retrospective study was approved by the institutional ethics committee of the hospital (Research Ethics Committee of the second hospital of Hebei Medical University. Approval Letter No.2019-P037) performing the study, all procedures were performed in compliance with the 1975 Declaration of Helsinki tenets and its later amendments. All patients were fully anonymized before we accessed them. All patients signed the informed consent for MRI safety examination. The study cohorts enrolled in this research were according with the criteria as follows: ①Surgical and pathological diagnosis of glioma in PACS system; ②All patients underwent scan of MR T2 FLAIR sequence before surgical; ③Available Ki-67, S-100, vimentin and CD34 immunohistochemical data; ④No intracranial decompression, chemotherapy or radiotherapy were performed before MRI scanning; ⑤Sufficient image quality without head movements or other artifacts. A total of 51 confirmed glioma patients between March 2015 to June 2018 from the hospital were screened in the end (30 males and 21 females, average age: 48.82 ±13.36 years, range: 7~72 years). Immunohistochemistry were not available for all, there were only 50 Ki-67 immunohistochemistry cases, 43 S-100 immunohistochemistry cases, 44

vimentin and 42 CD34, respectively. The clinical manifestations included headache, dizziness, slurred speech and limb weakness. More clinical information is listed in S1 Table.

## MRI protocol

MRI were all performed using a 3.0-Tesla scanner Achieva (Philips, Nederland) with 8-channel head coil. T2 FLAIR scanning parameters were as follows: TR = 8000~9000ms, TE = 125~140ms, FOV = 230mm×219mm, slice thickness = 6mm, slice gap = 6mm, matrix = 232×181. All MR image data come from a single center MR scanner.

## Immunohistochemical status

All the patients involved in the current study have received the operative resection of gliomas tumors. And the immunohistochemical analysis of the four biomarkers including Ki-67, vimentin, S-100 and CD34 were conducted based on the resected gliomas tumors. The expression of Ki-67, vimentin, S-100 and CD34 was detected by SP staining using BenchMark GX automatic immunohistochemical apparatus. We used rabbit anti-human monoclonal antibody for determination of Ki-67 (product number: RMA-0542) and vimentin (product number: RMA-0547) expression, used mouse anti-human monoclonal antibody for determination of CD34 (product number: Kit-0004) and S-100(product number: Kit-0007) expression. These preparations were all from Maixin Biotechnology Development Co., Ltd. The positive Ki-67 protein expression was defined by the brownish yellow stained nucleus. Vimentin positive expression was defined as brown or yellow stained nucleus or cytoplasm. S-100 protein positive expression was localized to brown cytoplasm and nuclear staining. CD34 staining was positively located as vascular endothelial cell membrane or cytoplasm was brownish yellow. Ki-67 labelling index (Ki-67 LI) was defined as the percentage of positive tumor cells to the total number of tumor cells in randomly selected 10 fields of magnificent view (×400). According to Beesley et al [37], Ki-67 LI was divided into 4 levels according to the positive rate: 0–5% was level 0, 6–25% was level 1, 26–50% was level 2, and more than 50% was level 3. In this study, the Ki-67 cohort was divided into label 0 group (level 0 and level 1, positive expression) and label 1 group (level 2 and level3, strongly positive expression). The S-100, vimentin and CD34 immunohistochemical indicators were divided into label 0 group (negative expression) and label 1 group (positive expression). Immunohistochemical staining results were evaluated by two experienced pathologists. The two pathologists independently reviewed all immunohistochemical results individually first and then reviewed together. Any discrepancies between two readers were discussed until a final consensus was generated.

## Radiomics feature

The entire technical flowchart of this study was briefly shown in Fig 1. All MR images were exported from PACS workstation as DICOM format. Before feature extraction, z-score standardization was applied to images. Since the data is single center and the scanning consistency is good, resampling and bias field correction are not adopted in this study. The 3-dimensional VOIs (3D VOIs) were manually segmented by an experienced radiologist (9 years of radiology experience) using ITK-SNAP software (version 3.8.0, www.itksnap.org). The VOIs that were used for feature extraction were specified as entire FLAIR abnormality that drew along the edge of edema on 2-dimensional axial map of each layer and automatically merged into 3D VOIs. More information about the lesions will be obtained from 3D VOIs such as the lesions' spatial distribution with surrounding tissues, tumor bulk and other heterogeneous features, etc.

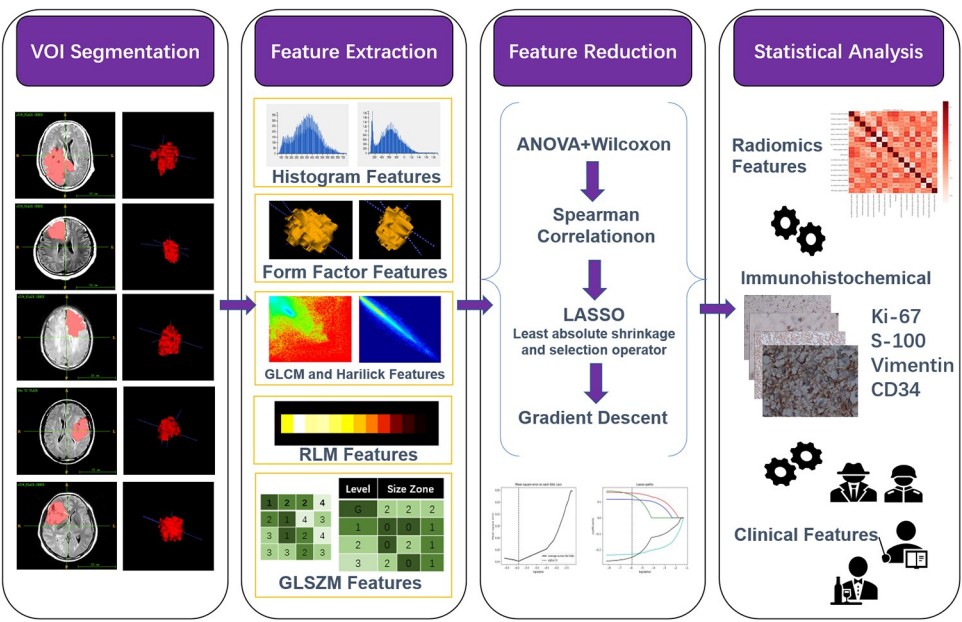

**Fig 1. An overview of the tech-flowchart of entire study. Step i**. MRI were imported into ITK to manually sketch VOI. **Step ii**. extract high-throughput radiomics features using AK software. **Step iii**. the feature set was performed redundancy reduction. **Step iv**. Ki-67, S-100, vimentin and CD34 immunohistochemical status were used as labels, and binary logistic regression models were established based on the selected features and significant clinical factors for non-invasive and accurate prediction of immunohistochemistry for providing a more personalized glioma management for different grades.

AK (Artificial Intelligence Kit, GE Healthcare, Shanghai China) software was used for features extraction, data preprocessing and feature reduction. AK software extracted a variety of high-throughput features according to the image biomarker standardization initiative (IBSI), including: ①Histogram features, mainly statistic the grey intensity or brightness information of the lesion; ②Form Factor features, mainly describe the shape and compactness of the lesions mathematically; ③Texture features, including gray level cooccurrence matrix (GLCM) features, Haralick parameters, gray level run length matrix(GLRLM) features and the Grey Level Size Zone Matrix (GLSZM) features. The GLCM features were calculated by counting the occurrence probability of the pixel pairs from different directions and with different steps. Haralick parameters were mainly based on the GLCM and calculated the sum-average of the features from four directions (0˚, 45˚, 90˚, 135˚) with the different offset 1. Therefore, the Haralick feature had directional invariance, that is, the Haralick eigenvalues will not change even if the original lesions rotate; The GLRLM features were obtained mainly by counting the probability of successive occurrence of pixels in different directions and steps; Gray level size zone matrix (GLSZM) features mainly by counting the number of pixels with the same adjacent gray value to get the Grey Level Size Zone Matrix. The offsets of GLCM and GLRLM features were 1, 4 and 7. The texture features were quantified finally to describe the complexity of the local lesions. The radiomics feature details were provided in the supplementary material(S2 Table).

Data preprocessing were applied: first, replace the abnormal value with median; second, standardize the data to eliminate the dimension effect. Feature reduction were performed by ANOVA + Mann-Whiney (variance analysis + U test), spearman correlation analysis and LASSO model. LASSO model choose the optimal log(α) according to the minimum mean square error of 10-folds cross validation to constructed a penalty function which could reduce

feature redundancy by compressing the coefficient of the unimportant features to zero (S1 Fig). If the feature cluster was still redundant, the GBDT algorithm was implemented for further reduction.

S-100 and vimentin immunohistochemical data adopted SMOTE over-sampling technology because of the serious bias of positive and negative distribution which can potentially improve the model efficacy [38–40].

### Binary logistic regression models

In this study, Ki-67, S-100, vimentin and CD34 immunohistochemical results were used as labels (label 0, label 1). Four feature clusters were finally screened out and four binary logistic regression classifiers were trained. In the current study, we only selected 51 patients with surgical and pathological diagnosis of gliomas, in which the immunohistochemistry of the four markers were not available for all. Therefore, we used and compared three kinds of validation methods including 3- and 5-fold cross validation and bootstrap, which are common approaches for model validation for small sample size [41,42](S3 Table).

For 3 or 5-folds repeat cross validation, all data were divided into 3 or 5 mutual exclusion subsets, 2 or 4 of which were used as training group data in turn and the remaining subset as validation data, then reselect different subsets as training and verification until all combinations of calendaring. Repeat above procedure 10 times. Additionally, bootstrap method repeat 100 times by sampling with return was also used. A total of 30, 50 or 100 accurate scores were obtained and the distribution (average, first quartile, third quartile) of score was used to evaluate the model performance so as to avoid model over-fitting. Clinical features significantly different between two labels were incorporated into the radiomics model. Hosmer-Lemeshow test was implemented for fit-goodness testing. AUC, sensitivity, specificity, accuracy and decision curves were used to evaluate the performance of models. DeLong test of AUC was used to compare the classification effectiveness between models.

### Statistical analysis

R studio (1.1.463, packages, "verification", "pROC", "rms", "glmnet", "caret" and "rmda", etc.) and IBM SPSS Statistics.22 were used for statistical analysis. The normal distribution (Shapiro test, $P>0.05$) and the homogeneity test of variance (Bartlett test, $P>0.05$) of the continuous variables were conducted. Distribution differences between two groups were analyzed by independent sample T test (when satisfying the normality test and homogeneity test of variance) and Kruskal-Wallis H test (when not satisfying the normality test and homogeneity test of variance). Pearson Chi-Square test or Fisher's Exact Test were performed for qualitative variables. $P<0.05$ were considered to be significantly different, however, $P\geq0.05$ but has practical clinic meaning were adopted in the model as well.

## Results

A total of 396 features were obtained from each case (S4 Table). There were no significant differences in age and gender between every two labels (all $P>0.05$) (Table 1). We proposed four comprehensive models in revealing immunohistochemical typing of Ki-67, S-100, vimentin and CD34. Ki-67 model composed of five features; S-100 model include five features and vimentin radiomics model enrolled three features, CD34 model composed of 3 features (Table 1). Form Factor features were not included in either the four models, GLCM and GLRLM were included in each model, and the ratio of them was relatively high in the corresponding feature clusters (Ki-67:60%; S-100: 80%; vimentin: 100%; CD34: 100%). Profiles of the selected four feature clusters can be found in S5, S6, S7 and S8 Tables. A correlation

**Table 1. Selected feature clusters of four models and the significant difference test of clinical features.**

| Immuno-histo-chemistry | Feature Cluster | | | | | P value | |
| --- | --- | --- | --- | --- | --- | --- | --- |
| | Histogram | GLCM | Haralick | GLRLM | GLSZM | sex | age |
| Ki-67 | kurtosis | ClusterProminence_AllDirection_offset1_SD<br>Inertia_AllDirection_offset1_SD | HaralickCorrelation_AllDirection_offset7 | — | SizeZoneVariability | 0.271 | 0.108 |
| S-100 | Min Intensity | ClusterProminence_AllDirection_offset1_SD<br>Correlation_angle90_offset1<br>GLCMEntropy_AllDirection_offset4_SD | — | LongRunLowGreyLevelEmphasis_angle45_offset1 | — | 0.582 | 0.366 |
| vimentin | — | GLCMEntropy_angle45_offset1 | — | ShortRunEmphasis_angle135_offset4<br>RunLengthNonuniformity_AllDirection_offset4_SD | — | 0.684 | 0.714 |
| CD34 | — | GreyLevelNonuniformity_angle135_offset4 | — | LowGreyLevelRunEmphasis_angle135_offset1<br>ShortRunEmphasis_angle135_offset7 | — | 0.179 | 0.388 |

Note: — Indicates that no such kind of feature was adopted in the model.

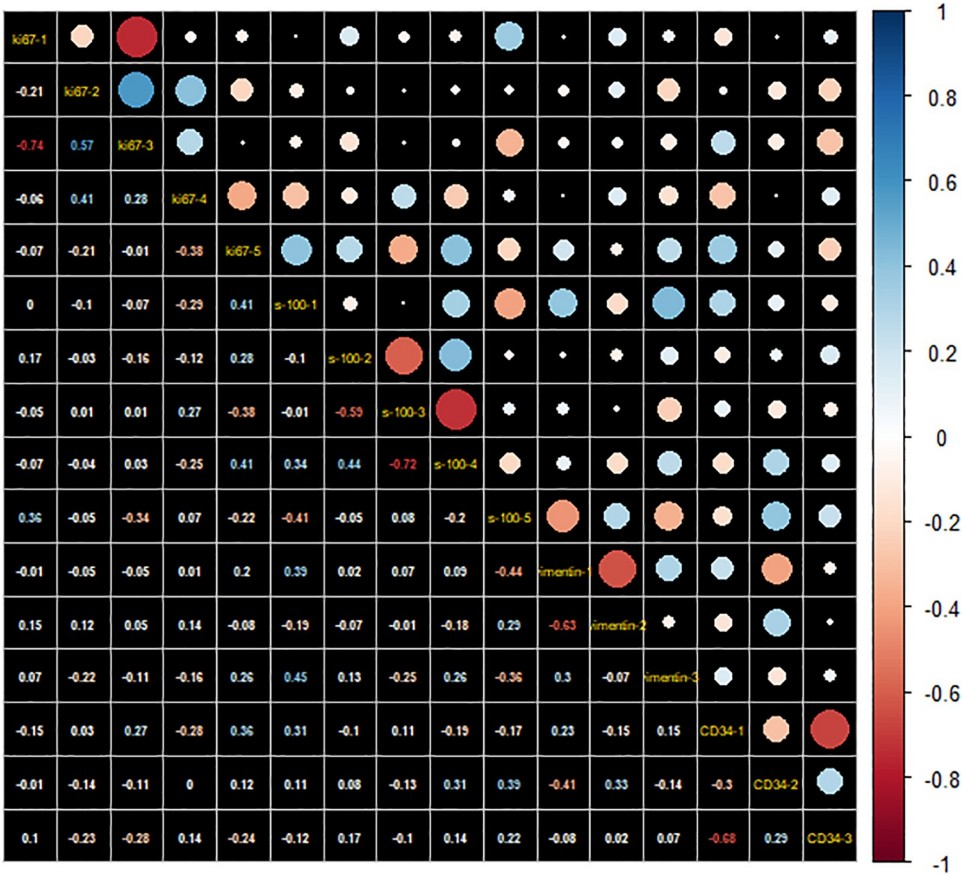

**Fig 2. Correlation matrix heatmap of selected radiomics features.** The correlation coefficient among the 16 features were basically low, suggesting that the four feature clusters were not redundant, and each feature contributed a unique information to the models. The magnitude of the correlation was indicated in the color bar on the right.

heatmap of the 16 radiomic features was shown in Fig 2, the low correlation coefficients between the 16 features indicated little redundancy among every feature cluster. It also suggested that the information and predictive effects provided by single radiomics feature was independent and unique.

Four discrimination radiomic signatures for predicting were established, as shown in S1, S2, S3 and S4 Formulas. The positive risk probability of each case can be obtained by S5 Formula.

In the different internal validation method, the results of ROC show that the range of value transformation under different validation methods is almost the same. In addition, the relatively small deviation between median and mean value in each validation process reflected the logistic regression model is stable for the current dataset. For example, as shown in S2 Fig. in the supplementary information, the ROC for model of Ki-67, vimentin and S-100 under bootstrap method is dense at a special range. While the two separate peaks for the ROC density of CD34 model indicated the existing instability. Nevertheless, the validation results initially indicated that logistic regression model based on radiomics features could potentially predict the expression of the pathological biomarkers.

The distribution of Radscore values of four models were shown in Fig 3. Radscore of each model were significantly different in two labels (all $P<0.05$, Kruskal-Wallis H test). In the Ki-67, S-100, vimentin and CD34 models, the median values of label 1 were larger than those of

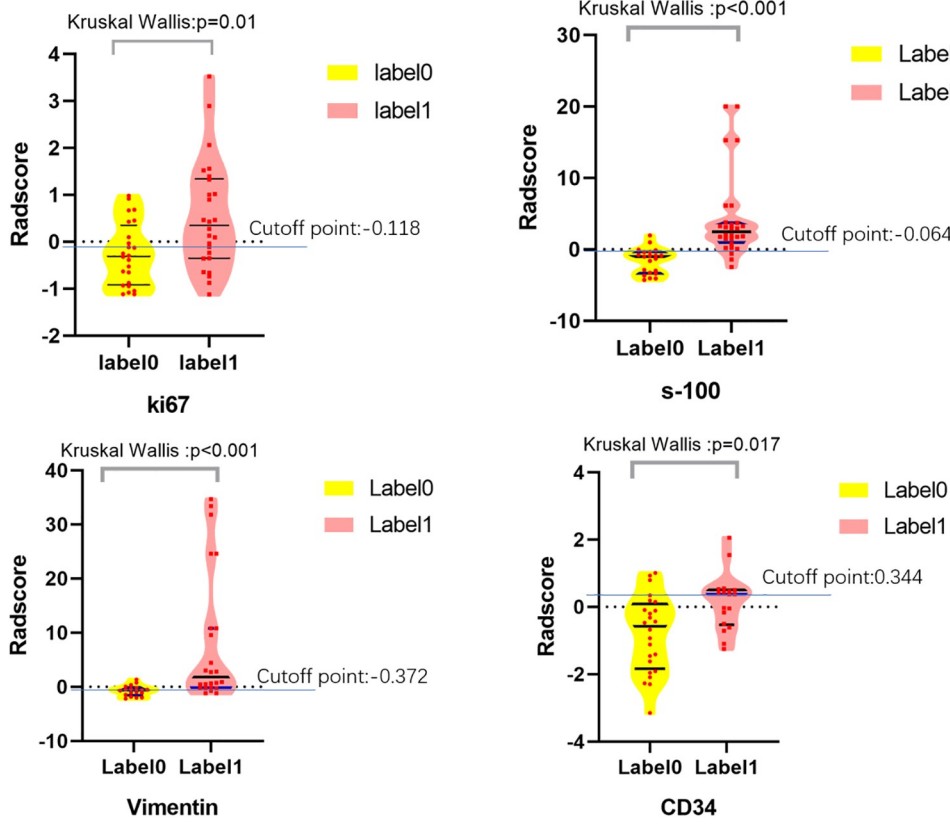

**Fig 3. The violin distribution of Radscore of four models between the two labels(A-D).** The yellow violin was negative protein expression group and the pink was positive group. The three solid lines in the violin represent the first quantile, the median and the fourth quantile of Radscore from bottom to top. The blue line was the cutoff value of Radscore. It can be seen from the figure that the Radscore values were significantly different between the two labels. The Radscore of the lable0 were generally smaller than label1. According to the cutoff value, the Radscore can be divided into a high-risk group and a low-risk group and guaranteed a high model prediction accuracy at the same time.

label 0. The first quartile of label 1 was larger than the fourth quartile of label 0 significantly in S-100 and vimentin models. The first quartile of label1 was less than the fourth quartile of label 0 in Ki-67 and CD34 models, but the median value was greater than the fourth quartile value. The Radscore values of each model were divided into high risk groups and low risk groups according to the cutoff values (Fig 3, Table 2). The Radscore cutoff value of Ki-67 was -0.118, Radscore greater than -0.118 were classified as Ki-67 strongly positive high-risk group, and the less than -0.118 was the positive high-risk group; the S-100 has a Radscore cut-off value of -0.064, ie, the Radscore greater than -0.064 were sorted in S-100 positive high-risk group, less than -0.064 was a positive low-risk group; vimentin's Radscore cut-off value was -0.372, that was, Radscore greater than -0.372 were as positive high-risk group, less than -0.372 were vimentin-positive low-risk group; CD34's cut-off Radscore was 0.344, the Radscore greater than 0.344 were highly-risk in being positive CD34 and the less than 0.344 in negative highly-risk group. The cutoff Radscore values guarantee the model to achieve both the maximum sensitivity and specificity at the same time. Therefore, the values of Radscore can be used as a significant factor in immunohistochemical classification in all four models.

Hosmer-Lemeshow tests were conducted for fit-goodness testing of four models. The$\chi^2$ values of Ki-67, S-100, vimentin and CD34 were 2.975, 2.489, 6.833 and 9.214, respectively,

**Table 2. The predictive performance evaluation parameters of four models.**

| Parameters | Ki-67 | S-100 | Vimentin | CD34 |
|---|---|---|---|---|
| Sample size | 50 | 43 | 44 | 42 |
| Positive group | 26 (52.00%) | 28 (57.14%) | 24 (57.14%) | 18 (42.86%) |
| Negative group | 24 (48.00%) | 21 (42.86%) | 18 (42.86%) | 24 (57.14%) |
| AUC±SE[a] (95%CI)[b] | 0.713±0.073 (0.568~0.832) | 0.92±0.0381 (0.811~0.980) | 0.854±0.0579 (0.711~0.944) | 0.745±0.077 (0.587~0.867) |
| Significance level P (Area = 0.5) | 0.0036 | <0.0001 | <0.0001 | 0.0014 |
| Youden index J | 0.3590 | 0.7976 | 0.5972 | 0.4306 |
| Associated criterion | >-0.118 | >-0.064 | >-0.372 | >0.344[c] |
| Sensitivity (95%CI) | 69.23(48.2~85.7) | 89.29(71.8~97.7) | 87.5(67.6~97.3) | 55.56(30.8~78.5) |
| Specificity (95%CI) | 66.67(44.7~84.4) | 90.48(69.6~98.8) | 72.22(46.5~90.3) | 87.50(67.6~97.3) |
| Accuracy | 66.0 | 0.898 | 0.738 | 0.667 |
| AIC | 72.509 | 46.163 | 45.037 | 56.654 |
| +LR[d](95%CI) | 2.08 (1.1~3.9) | 9.38(2.5~35.3) | 3.15(1.5~6.7) | 4.44(1.4~13.8) |
| -LR[e] (95%CI) | 0.46 (0.2~0.9) | 0.12(0.04~0.3) | 0.17(0.06~0.5) | 0.51(0.3~0.9) |
| +PV[f] (95%CI) | 69.2 (54.7~80.7) | 92.6(76.9~97.9) | 80.8(66.3~90.0) | 76.9(51.7~91.2) |
| -PV[g](95%CI) | 66.7 (51.3~79.2) | 86.4(68.3~94.9) | 81.3(59.1~92.8) | 72.4(60.5~81.8) |

[a] Standard Error.

[b] 95% Confidence interval, Binomial exact.

[c] Optimal criterion which was taking into account disease prevalence (42.9%).

[d] Positive likelihood ratio.

[e] Negative likelihood ratio.

[f] Positive predictive value.

[g] Negative predictive value.

*P* values were 0.936, 0.928, 0.555 and 0.325, respectively. Results showed that there were no significant differences between the four classification models and the corresponding actual models. Among them, the S-100 model and the actual model had the best fit-goodness. In addition, the Akaike information criterion (AIC) of Ki-67, S-100, vimentin and CD34 models were 72.509, 46.163, 45.037 and 56.654, respectively. The results showed that the fit-goodness of S-100 and vimentin models were better than that of Ki-67 and CD34 models again. However, the specific reasons for the relative unreliability of Ki-67 and CD34 models need to be further verified. In addition, the S-100 model had the highest positive likelihood (9.38) ratio and the smallest negative likelihood ratio (0.12), indicating that the probability of the correct judgement using model when predict the positive and negative expression of S-100 protein was much greater than the wrong judgment. Both higher positive predictive values (92.6) and negative predictive values (82.4) also indicate higher accuracy for S-100 model predictions. The high predictive performance was followed by vimentin model. However, in the Ki-67 and CD34 models, the prediction performance were relatively poor in terms of comprehensive indicators (Table 2).

The ROC curves were shown in Fig 4. The results showed that the S-100 and vimentin models had good classification performance, while the Ki-67 and CD34 models were poorly behaved, the sensitivity of CD34 was only 55.56% (Table 2), which suggests that CD34 model based on the data in this study had no effective predicting performance. In Fig 4, DeLong-test of AUC demonstrated that the performance of S-100 model was significantly greater than that

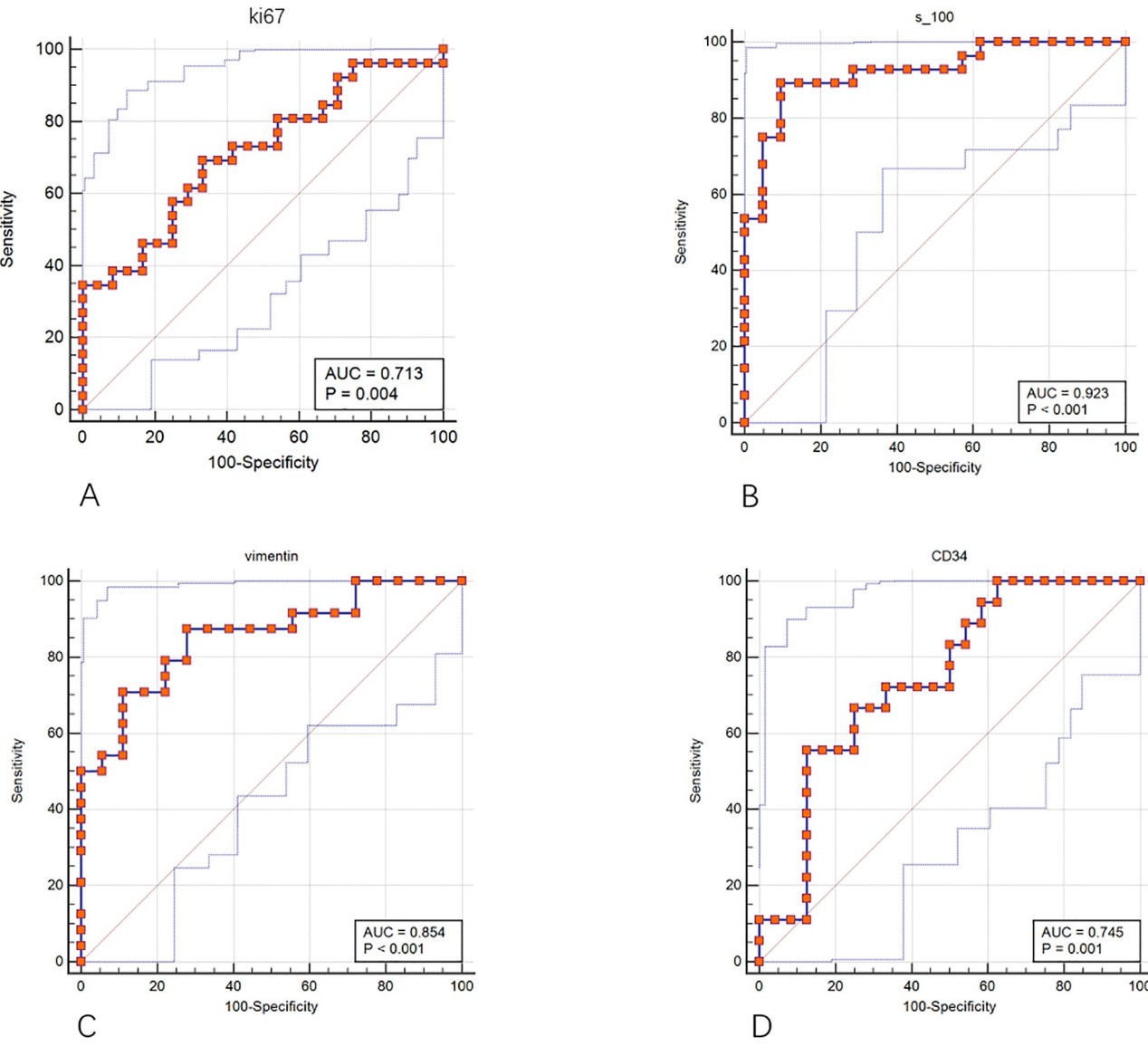

**Fig 4. ROC curves of four models for predicting the immunohistochemical subtype.** (A)The ROC curve of Ki-67. (B)The ROC curve of S-100. (C) The ROC curve of vimentin. (D)The ROC curve of CD34. The red dots connecting lines were ROC curves, and the two blue thin lines were 95% confidence intervals. S-100 model has the largest AUC prediction.

of Ki-67 ($P = 0.013$) and CD34 ($P = 0.043$) models, while there were no significant differences between other two models (all $P>0.05$). The calibration curves of the four models were shown in Fig 5. The classification effects of the models were shown in Fig 6. It can be seen that the Ki -67 and CD34 models were slightly inferior performed to the other two models.

The decision curves of the four models were displayed in Fig 7. It can be seen from the graph that when the probability threshold (pt) outweighed 0.25, the net benefit of Ki-67 model was the largest compare with the all treatment and none treatment; Similarly, the pt of the S-100 model was greater than 0.01, vimentin was more than 0.01. The net benefit of CD34 model was less than that of all treatments within a wide threshold, it can achieve maximum net benefit only between 0.46~0.62 and 0.75~0.90.

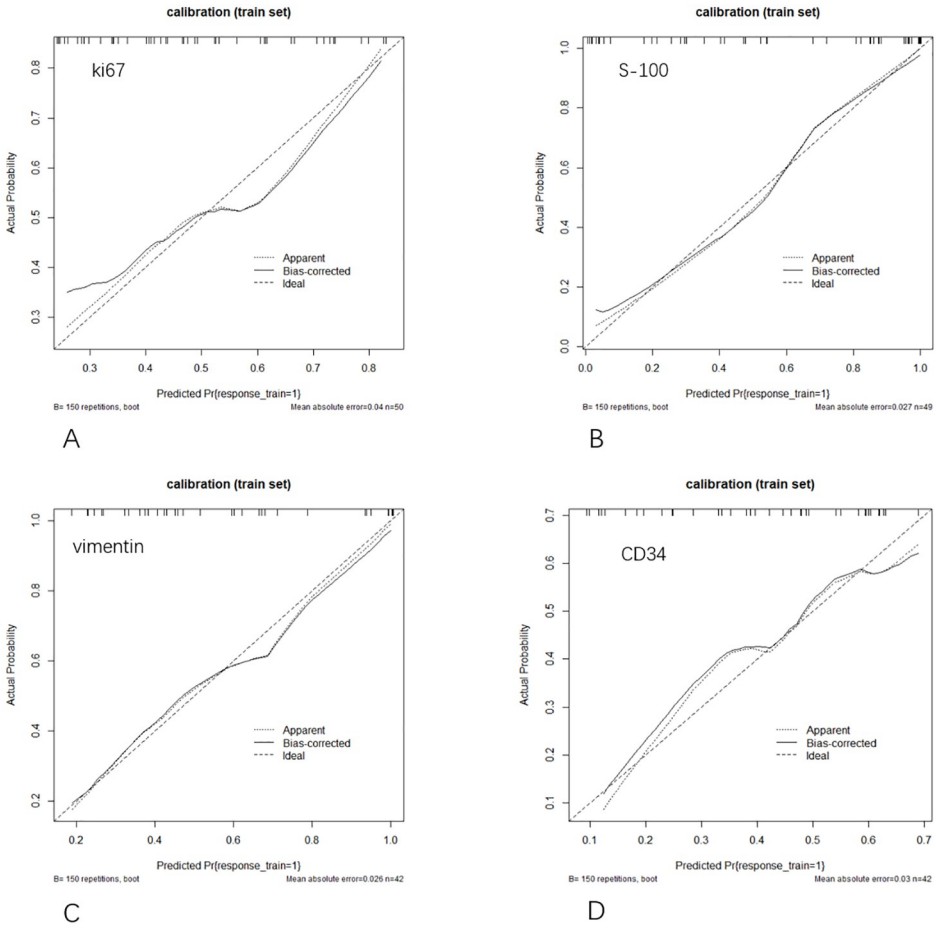

**Fig 5. Calibration curves of four models for predicting the immunohistochemical subtype.** (A) Calibration curve of Ki-67 model. (B) Calibration curve of S-100 model. (C) Calibration curve of vimentin model. (D) Calibration curve of CD34 model. Diagonal dotted line marks the location of the ideal model. The dotted line represents the predicted performance of the model, and the solid line was the bias correction in the model.

## Radiomics features are significantly correlated with glioma grade

In addition, we used the same 51 cohort to classify the high and low grades of gliomas, and found that the radiomics features were significantly correlated with glioma grade (Fig 8). The case **i** in the graph was glioma grade IV, and cases ii and iii were glioma grade II. The GLCM map and the RLM map of the two different grades were significantly different. Four features were included in the identification model finally, they were two GLCM features (Haralick Correlation_angle135_offset7 and Inverse Difference Moment_AllDirection_offset4_SD) and two GLRLM features (LowGrey Level RunEmphasis_AllDirection_offset1_SD and hortRunEmphasis_Direction_offset7), The proportion of GLCM and GLRLM in feature clusters was 100%. The chi-square value of fit -goodness in this model was 2.797, $P = 0.946$, AUC: 0.888, sensitivity: 0.781, specificity: 0.895. The calibration parameters were mean absolute error = 0.023, quantile of absolute error = 0.049. In addition, we found that the age was normal distribution (Shapiro-Wilk test, $P = 0.2353$), and were statistically different between the high and low grades of glioma ($P = 0.015$), while sexes had no statistical difference between the two groups ($P = 0.489$). Combining age and radiomics features could significantly improve the

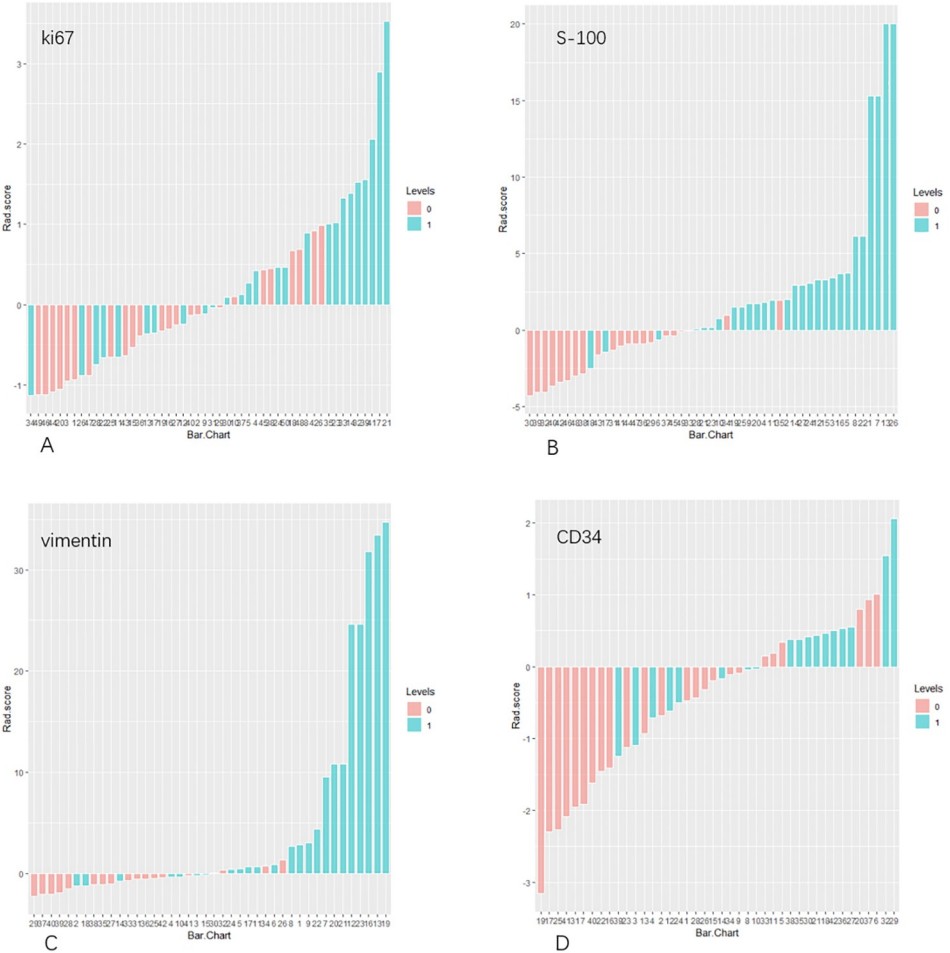

**Fig 6. The identification effect of the four predicting models.** (A) identification effect of Ki-67 model. (B) identification effect of S-100 model. (C) identification effect of vimentin model. (D) identification effect of CD34 model. Pink represents the gold standard negative group and blue represents the gold standard positive group. Pinks greater than 0 and blues less than 0 were cases incorrectly identified by model.

model performance. The Chi-square value of fit-goodness of this model was 3.477, $P = 0.901$, AUC: 0.929, sensitivity: 0.938, specificity: 0.789. Additionally, the calibration parameters were mean absolute error = 0.028, quantile of absolute error = 0.061. The ROC curve, calibration curve and identification effect diagram were shown in S3, S4 and S5 Figs.

## Discussion

We screened out four sets of high-order radiomics feature clusters based on T2 FLAIR through supervised machine learning and established four predictive models for immunohistochemical biomarker prediction using the positive/negative pathological results of Ki-67, S-100, vimentin and CD34 as labels in gliomas patients. The results showed that the high-order radiomics features are potential predictors. The cohort enrolled in this research can be modeling for predicting the gliomas grades as well. Therefore, the classifier based on radiomics features can provide a noninvasive and personalized management method for glioma patients.

Since the Dutch scholar Lambin et al [43] proposed radiomics in 2012, high throughput textures analysis as a new technology has been applied to the studies of tumors diagnosis,

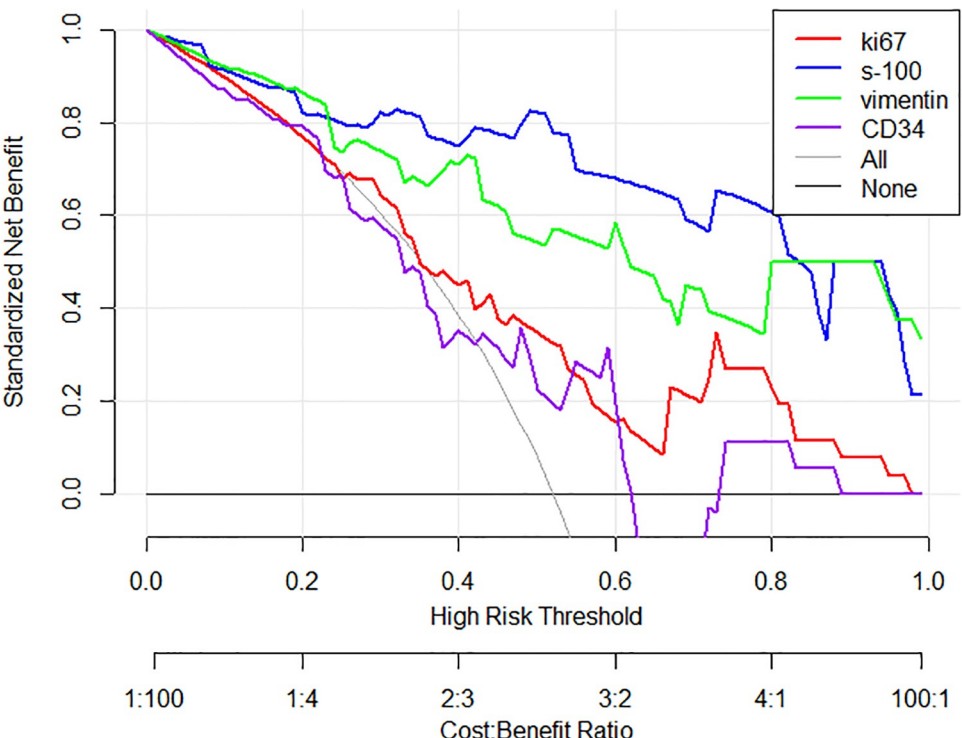

**Fig 7. The decision curves of the four models.** The thin gray line represents the treatment for all patients, and the horizontal thick gray line means that all patients were not treated. The Y-axis represents the standard net benefit generated using the corresponding model, the X-axis represents the threshold range and the scale axis below the x-axis was the cost-benefit ratio at the corresponding threshold probability. It can be seen that within a relatively large threshold range, when the Ki-67, S-100 and vimentin models were used for decision-making, the standard net benefit was greater than that of treating all patients or treating none patients. The CD34 model had weak reference value except for threshold probability between 0.75–0.9.

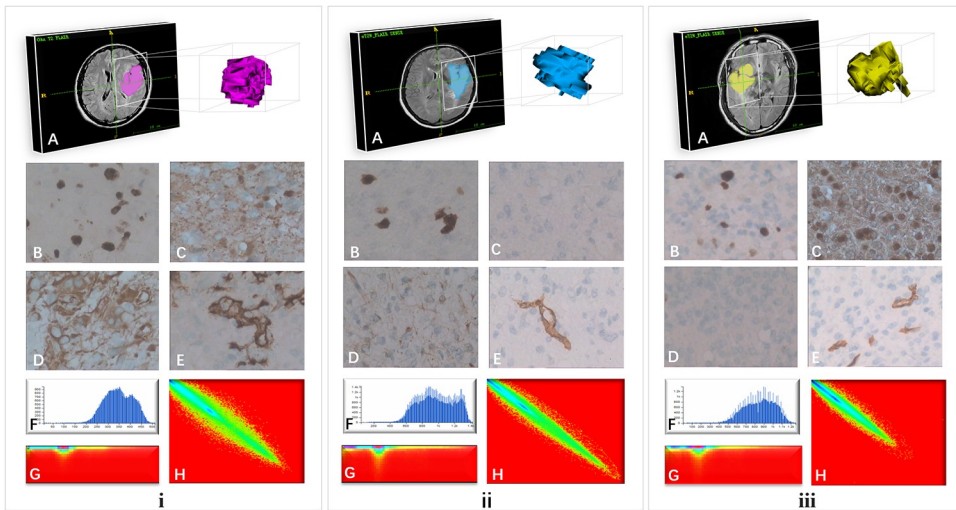

**Fig 8. Immunohistochemistry and radiomics features of three cases. i** Male, 52 years old, WHO grade IV, Ki-67 (50%), S-100, vimentin, CD34 were positive expression. **ii** Female, 43 years old, WHO grade II, Ki-67 (10%), S-100, CD34 negative expression, vimentin positive expression. **iii** Male, 43 years old, WHO grade II, Ki-67 (8%), vimentin were negative expression, S-100, CD34 were positive expression. Among them, **A** VOI of the case; **B** Ki-67; **C** S-100; **D** vimentin; **E** CD34; **F** histogram of VOI; **G** RLM of VOI; **H** GLCM of VOI.

staging, grading and predictive treatment more extensively. Previous studies mainly focused on the glioma grading [44,45]. Nevertheless, only glioma grading cannot satisfy clinical demands any more, the same WHO grading may show different prognosis due to the different pathological types in oncology genetics. In our study, the second and third cases in Fig 8 were both glioma grade II. Although pathological confirmed same grade, their immunohistochemical expressions were extremely different. In the second case, the expression of S-100 and CD34 was negative, but in the third case, the expression of S-100 and CD34 was positive; the expression of vimentin was positive in the second case while the negative in the third case. Ki-67 expression was positive in both cases. Therefore, fully considering the molecular pathology may greatly help confirming gliomas heterogenous sub-type and prognostic prediction. Exploring more biomarkers related to gliomas molecular typing, and the association of imaging-based radiomics features with these biomarkers is conducive to realize easy-to-implement radiomics-assisted multi-biomarker "histological" analysis and promote the personalized treatment of gliomas. However, the first step is to validate if the radiomics is effective in identifying the molecular biomarkers. So in the current study, we try to use radiomics features to initially identify four markers of Ki-67, vimentin, S-100 and CD34, which are histological markers recommended in several gliomas diagnosis [9,10,20,46]. In the Ki-67, S-100, vimentin and CD34 radiomics models established in this study, GLCM and RLM were significant predictive features, their radiomics map showed significantly different distribution in Fig 8. The Ki-67 expression of case i was different from case ii and iii, and the GLCM and RLM also showed differences. These indicated that the models established in this study are potential for predicting the positive or negative expression of immunohistochemical biomarkers. It might utilized as a supplementary method for traditional pathological biopsy which has limitations such as invasiveness, sampling error and difficulty in obtaining satisfactory data during clinical practice [47]. Moreover, there is a lack of studies on the differentiation of negative and positive pathological biomarkers by T2 FLAIR radiomics at present. In this study, radiomics prediction models for Ki-67, S-100, vimentin and CD34 biomarkers were established based on T2 FLAIR images.

First, high-throughput radiomics features were de-redundant to reduce the complexity of the model to improve the model performance. Four effective feature clusters were screened out. We found that the proportion of higher-order GLCM and RLM features in each feature cluster were rather high (Ki-67: 60%; S-100: 80%; vimentin: 100%; CD34: 100%; high and low grade gliomas: 100%), which indicating that radiomics features were more effective than the traditional image morphology features. Radiomics features contributed more for improving the accuracy of the model, which was consistent with previous research [48–53]. The correlation analysis of the total selected 16 features showed low feature redundancy in each model and each feature have provided independent prediction information. Then we established four sets of Radscore and found that the Radscore was significant factor for predicting four immunohistochemical biomarkers. The best prediction performance was observed in the S-100 model (AUC: 0.920; sensitivity: 89.29%; specificity: 90.48%; accuracy: 82.00%). S-100 proteins are involved in tumor occurrence and development including regulation of cell differentiation, cell cycle progression, cell proliferation, cell apoptosis, cell motility, invasion and migration, tumor microenvironment and cancer stem cells (CSCs) [54]. Several studies revealed that S-100 protein members were related to gliomas subtype identification, tumor progression and therapy effectiveness. Camby et al [55] indicated that several S-100 proteins could help differential diagnosis or judge malignancy of human astrocytic tumors. For example, modifications in the level of S100A3 protein expression level could help identify the pilocytic astrocytomas from WHO grade II-IV astrocytic tumors. While S100A6 protein expression enabled a clear distinction between low (WHO grade I and II) and high (WHO grade III and IV) grade

astrocytic tumors. S100A1, S100A4 and S100B protein expression are related to the increasing levels of tumor malignancy. Another study reported that S100A4 played a crucial role in neu-trophil-promoting tumor progression and S100A4 depletion could increase the effectiveness of anti-VEGF therapy in glioma [56]. S-100 also might be valuable in prognostication of sur-vival for glioma patients [57,58]. The good S-100 prediction performance of T2 FLAIR radio-mics in the current study implies its potential in the future applications in gliomas treatment decision-making and effectiveness assessment.

The vimentin model (AUC: 0.854; sensitivity: 87.50%; specificity: 72.22%; accuracy: 73.80%) had good prediction performance as well. The overexpression of vimentin has been demon-strated to be associated with an increased migratory or invasive capacity of the cancer cells [59]. Therefore, vimentin is considered as a potential indicator for cancer prognosis and thera-peutic target [60]. Vimentin was postulated as a molecular marker presenting enhanced motil-ity and invasion in gliomas [17,18,61]. Its expression indicates a lower degree of differentiation and was found in high-grade astrocytomas [46]. As vimentin indicated enhanced invasion and cells of invasive gliomas show a decreased proliferation rate and a relative resistance to apopto-sis [18], high vimentin expression could be taken as a prognostic factor for treatment difficulty or poor survival in diffusive glioma or high-grade glioma patients. Lin et al [12] systemically analyzed the vimentin expression and found that vimentin expression was associated with tumor grade and overall survival of high-grade glioma patients. High vimentin expression was an independent significant prognostic factor for poor survival in high-grade glioma patients while low vimentin expression a biological indicator of better response to temozolomide ther-apy for glioblastoma patients [19]. It is encouraging that T2 FLAIR radiomics could be further explored to validate its usefulness in imaging-based assessment of migratory, invasion and related prognostication for different gliomas subtypes.

The Ki-67 model exhibited moderate performance (AUC: 0.713; sensitivity: 69.23%; speci-ficity: 66.67%; accuracy: 66.00%). The preoperative prediction of Ki-67 LI may help gliomas grading and prognosis prediction which are both important factors during treatment deci-sion-making. It is increasingly appearing studies which try to explore relationship between MRI features and Ki-67 expression. Gates et al [14] demonstrated that the Ki-67 LI was corre-lated with MRI conventional features and functional parameters (T2-weighted, fractional anisotropy, cerebral blood flow, Ktrans), which could be used to guide biopsy, resection or radiation in the glioma patients. Su et al [11] found that multi-contrast MRI radiomics were significantly correlated with tumor grade and Ki-67 and provided independent but supple-mental information in assessing glioma proliferation behavior. In our results, although the comprehensive Ki-67 prediction efficiency of T2 FLAIR radiomics is moderate, it might intro-duce additional power for prediction of tumor proliferative behavior when combined with other conventional or functional MRI series. The CD34 model(AUC, 0.745; sensitivity, 55.56%; specificity, 87.50%; accuracy, 82.00%), had lower sensitivity while better specificity and accuracy. The use of CD34 for the prognosis, diagnosis, and treatment of various cancers has been increasing. The precise identification of CD34 noninvasively could help predicting angiogenesis-related gliomas progression [5–8]. Beside grading, CD34 expression might of assistance to indicate glioblastoma stem-like cells differentiation into tumor-associated endo-thelial cells in low-grade gliomas [62]. In addition, CD34 might have strong potential in pre-dicting gliomas survival and therapy effectiveness [20]. CD34 staining were also associated with hypoxia-induced angiogenesis and may play a role in glioblastoma hemorrhage. Its underlying mechanism of which may promote the development individualized therapies for glioblastoma [63]. Therefore, our results indicate that T2 FLAIR radiomics might provide a noninvasive identifier of angiogenesis, which is useful for gliomas progression or prognosis analysis. However, the low sensitivity of the current model may need further consideration

and involving larger samples. In this study, we do not recommend comparing among the four models due to inconsistent sample data and was contrary to the purpose and principle of our research, the absolute prediction performance of the model was the goal. According to the standard net benefit of the decision-making curve, Ki-67, S-100 and vimentin models were proposed for immunohistochemical biomarker prediction. While the CD34 model was weaker for positive and negative discrimination, which may be related to the data distribution included in this study.

There are still some limitations in this study. Firstly, the sample size in this study was small, and all data were used as the training set of the model with only internal validating. Secondly, not all the immunohistochemical data were available of the 51 patients, the absolute evaluation index of the models may be affected by the different sample size. Finally, the aim of this study is to use immunohistochemical predictors to guide the management and predict prognosis of gliomas of different grades, but we have no follow-up data to stand by the significance of this study. In conclusion, further research is needed to explore the relationship between radiomics features and immunohistochemical indicators and establish more generalized prediction models. The more detailed relationship between radiomics features and protein subtypes, such as different S-100 members could be studied to reveal their predictive effectiveness for diagnosis or prognosis. Further research requires a large number of samples, complete immunohistochemical data and follow-up data to verify the significance of immunohistochemical biomarker identification based on radiomics.

## Conclusions

In summary, we have established four predicting models for four kinds of immunohistochemical biomarkers (Ki-67, S-100, vimentin and CD34) in gliomas patients based on T2 FLAIR radiomics, which are potential for imaging-based prediction of tumor proliferation, malignancy, therapeutic effectiveness, migratory or invasion, and angiogenesis. The S-100 and vimentin models have higher reliability and can effectively predict the expression of respective proteins. However, the comprehensive prediction efficiency of Ki-67, CD34 model are relatively low, the reasons need to be further explored. On the basis of glioma grades, it is expected to provide an intelligent, non-invasive and personalized assistant diagnosing tool for pathology of different gliomas subtypes.

## Supporting information

**S1 Fig. The example visualization process of feature deduplication of LASSO model.** (A) The lasso model first choose the optimal log (α) according to the minimum mean square error among the 10-fold cross validation.(B) The lasso model compresses the unimportant feature coefficients to zero according to the optimal log (α) value.
(TIF)

**S2 Fig. ROC density for each model under bootstrap method.**
(TIF)

**S3 Fig. The ROC curves of radiomics features and age and radiomics features combined model for differentiating high and low grade glioma.**
(TIF)

**S4 Fig. The calibration curves of radiomics features model and age and radiomics features combined model for differentiating high and low grade glioma.**
(TIF)

**S5 Fig. The identification effect of radiomics features model and age and radiomics features combined model for differentiating high and low grade glioma.**
(TIF)

**S1 Formula. Discrimination radiomic signatures for predicting of Ki-67.**
(DOCX)

**S2 Formula. Discrimination radiomic signatures for predicting of S-100.**
(DOCX)

**S3 Formula. Discrimination radiomic signatures for predicting of vimentin.**
(DOCX)

**S4 Formula. Discrimination radiomic signatures for predicting of CD34.**
(DOCX)

**S5 Formula. The positive risk probability of each case.**
(DOCX)

**S1 Table. Clinical information of patients enrolled in the study cohort.**
(XLSX)

**S2 Table. The primary radiomics features extracted in this study.**
(DOCX)

**S3 Table. The accurate score of Bootstrap validation method, 3-fold cross validation method and 5-fold cross validation method.**
(DOCX)

**S4 Table. 396 features extracted from each case.**
(CSV)

**S5 Table. The selected feature cluster of Ki67 model.**
(CSV)

**S6 Table. The selected feature cluster of S-100 model.**
(CSV)

**S7 Table. The selected feature cluster of vimentin model.**
(CSV)

**S8 Table. The selected feature cluster of CD34 model.**
(CSV)

**S1 File. Ethics documentation.**
(PDF)

**S2 File. Texture paramters.**
(PDF)

**S3 File. Statistical analysis.**
(ZIP)

## Acknowledgments

We acknowledge the Department of Pathology of The Second Hospital of Hebei Medical University, and Fuli Wang' guidance on pathology, we also appreciate Jialiang Ren' help and supporting from GE Healthcare.

## Author Contributions

**Conceptualization:** Jing Li, Siyun Liu, Ying Qin, Yan Zhang, Ning Wang, Huaijun Liu.

**Data curation:** Jing Li, Siyun Liu, Ying Qin, Yan Zhang, Ning Wang, Huaijun Liu.

**Formal analysis:** Jing Li, Siyun Liu, Ying Qin, Yan Zhang, Ning Wang, Huaijun Liu.

**Investigation:** Jing Li, Yan Zhang, Ning Wang, Huaijun Liu.

**Methodology:** Jing Li, Siyun Liu, Ying Qin, Yan Zhang, Ning Wang, Huaijun Liu.

**Project administration:** Jing Li, Huaijun Liu.

**Resources:** Jing Li, Huaijun Liu.

**Software:** Jing Li, Siyun Liu, Ying Qin, Huaijun Liu.

**Supervision:** Jing Li, Huaijun Liu.

**Validation:** Jing Li, Siyun Liu, Ying Qin, Huaijun Liu.

**Visualization:** Jing Li, Huaijun Liu.

**Writing – original draft:** Jing Li, Siyun Liu, Ying Qin, Yan Zhang, Ning Wang, Huaijun Liu.

**Writing – review & editing:** Jing Li, Siyun Liu, Ying Qin, Yan Zhang, Ning Wang, Huaijun Liu.

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
