## [Decision Letter · Decision Letter 0]

23 Oct 2019

PONE-D-19-22703

High-order radiomics features based on T2 FLAIR MRI predict glioma immunohistochemical subtype: a more precise and personalized gliomas management

PLOS ONE

Dear Dr. Liu,

Thank you for submitting your manuscript to PLOS ONE. After careful consideration, we feel that it has merit but does not fully meet PLOS ONE’s publication criteria as it currently stands. Therefore, we invite you to submit a revised version of the manuscript that addresses the points raised during the review process.

We would appreciate receiving your revised manuscript by Dec 07 2019 11:59PM. To enhance the reproducibility of your results, we recommend that if applicable you deposit your laboratory protocols in protocols.io, where a protocol can be assigned its own identifier (DOI) such that it can be cited independently in the future. For instructions see: http://journals.plos.org/plosone/s/submission-guidelines#loc-laboratory-protocols

We look forward to receiving your revised manuscript.

Kind regards,

Alessandro Weisz

Academic Editor

PLOS ONE

**Journal Requirements:**

2. In the ethics statement in the manuscript and in the online submission form, please provide additional information about the patient records used in your retrospective study. Specifically, please ensure that you have discussed whether all data and tissue samples were fully anonymized before you accessed them and/or whether the IRB or ethics committee waived the requirement for informed consent. If patients provided informed written consent to have data from their medical records used in research, please include this information in your ethics statement in the Methods section of the manuscript and the online submission form.

**Additional Editor Comments (if provided):**

The manuscript was evaluated by two Reviewers experts in this field that both found the study described of value and interesting, but still requiring major revisions before being further considered for publication.

**Comments to the Author**

1. Is the manuscript technically sound, and do the data support the conclusions?

Reviewer #1: Partly

Reviewer #2: No

2. Has the statistical analysis been performed appropriately and rigorously? 

Reviewer #1: No

Reviewer #2: Yes

3. Have the authors made all data underlying the findings in their manuscript fully available?

Reviewer #1: Yes

Reviewer #2: Yes

4. Is the manuscript presented in an intelligible fashion and written in standard English?

Reviewer #1: No

Reviewer #2: Yes

5. Review Comments to the Author

Reviewer #1: The authors propose a solution to a relevant clinical problem: intratumoral heterogeneity of glioma limits treatment choice and efficacy greatly, and radiomics indeed promises a relevant method to potentially tackle this problem as is stated by the authors and the references they refer to. MR-FLAIR can indeed be a very relevant sequence to use in this setting, especially with regards to invasiveness of the tumor. The study conducted is relevant for the field and quantitative MR imaging is currently very much studied within neuro-oncology. The manuscript could therefore be well suited for publication, however, there are some major remarks to be made at the manuscript in its current form:

- IHC analysis: the authors do not describe which antibodies are used for determination of protein expression, this should be mentioned for replication purposes. It is stated that this was analysed by two neuro-pathologists, however it is not explained what happens upon disagreement between these to pathologists.

- Feature extraction: the ROIs that are used for feature extraction are not specified by the authors (tumor? edema? entire FLAIR abnormality?).

- Statistical analysis: the authors only use their patient cohort to train radiomic feature models on. It would be recommended for the authors to look into validation methods for their models. It is understandable that external validation data would be difficult to come by but there are internal validation methods available which would provide more (reliable) data on the predictive power of these models. At this point, especially since the MRI sequences are obtained from one insititute and one scanner, it could very well be that the models are only applicable for this specific institute. After internal validation this could of course still be the case but this would improve the quality of the results and interpretation. The results provided present the possibility of radiomics models to predict IHC expression, however some kind of validation would be highly recommended to strengthen the manuscript and the conclusions of the authors.

- Patient cohort: did all patients included in this study receive a resection of their tumor? Often, glioma patients only receive a biopsy due to location of the tumor or physical condition of the patient. This could influence the IHC results obtained. Especially since the authors pose sampling-bias as a limitation of IHC analysis and radiomics as a solution to this it should be addressed what the IHC results are based upon.

- Language: This is a major concern for publication. The manuscript contains a lot of grammatical and spelling errors, this makes parts of the manuscript hard to read and fully comprehend. Without optimization of the use of language this manuscript is not suitable for publication. The authors have an incoherent use of abbreviations (some are spelled out multiple times, abbreviation is sometimes used and sometimes not and different abbreviations are used i.e. VOI/ROI). It is recommended to review this manuscript on the correct use of English, potentially by an external partner or language center before publication.

- Clinical relevance: the authors state that they want to identify immunohistochemical subtypes to aid precision medicine. However, the manuscript insufficiently states the clinical consequences of identifying these specific markers non-invasively. How can identification of these markers aid in the treatment plan that is clinically applied? It would be very helpful if the authors could provide data on the prognostic relevance of these IHC markers in their patient cohort to see whether identification of these markers can help in predicting patient prognosis.

In the discussion the authors correctly refer to the new WHO classification of CNS tumors which incorporate molecular markers but do not clarify what the relevance of this is compared to their work and vice versa.

Reviewer #2: General: This work proposes to use T2 FLAIR radiomic features in predicting the immunohistochemical subtype of glioma like Ki-67, S-100, vimentin and CD34. This work uses only limited dataset of 51 glioma patient samples which is not enough to validate the findings. Here some comments:

Abstract: Author mentioned the “The deep radiomics” this is mean that the authors used deep learning which is not the case in this paper. I suggest to use only the radiomics.

Introduction:

-The authors did a good state of the art regarding the immunohistochemical subtype. However, we suggest the authors to involve a section about radiomics for brain tumor and involve the following citations:

1-Integration of Radiomic and Multi-omic Analyses Predicts Survival of Newly Diagnosed IDH1 Wild-Type Glioblastoma

2-Predicting the gene status and survival outcome of lower grade glioma patients with multimodal MRI features

3-Radiomics in glioblastoma: current status and challenges facing clinical implementation

In addition, showing the differences with these literatures and how the proposed work is different.

Materials and methods:

- how the authors apply the normalization? We suggest authors to clarify all the preprocessing steps (normalization, resolution, field corrections…etc.).

-please clarify the number of samples for each of the labels to estimate the % for class 1 and class 2

-How the authors compute these radiomic functions more detail is recommended.

-Why the author chooses the logistic regression for classification and no other classifier model like random forest, SVM, CNN?

-I suggest the authors to apply multivariate analysis by dividing the patient’s data to 75 % training and 25 % testing and measure the performance metrics to avoid the doubt about over fitting.

In discussion: I suggest authors to avoid the first proposed radiomics in “Since the Dutch scholar Lambin et al [36] first proposed radiomics in 2012，high throughput textures analysis as a new technology has been applied to the studies of tumors diagnosis, staging, grading and predictive treatment more extensively.” Because there are many papers published before 2012 using the term of features extraction for predicting the cancer grade..etc.

6. PLOS authors have the option to publish the peer review history of their article (what does this mean?). If published, this will include your full peer review and any attached files.

Reviewer #1: No

Reviewer #2: Yes: Ahmad Chaddad

---

## [Author Response · Author response to Decision Letter 0]

23 Dec 2019

PONE-D-19-22703

High-order radiomics features based on T2 FLAIR MRI predict glioma immunohistochemical subtype: a more precise and personalized gliomas management

PLOS ONE

Response letter to reviewers

We are very grateful for each of the reviewers and editors of this manuscript for the insightful comments and suggestions, which greatly help us review our article and make revisions. The responses to reviewers are as follows.

NOTE: The citation number of the references throughout the response letter is accorded with the citation number in the manuscript. The references which were not cited in the manuscript were quoted as plain text as [authors, title, journal, year, volume, page or PMID or DOI]. In addition, we provide additional information about the patient records in the supporting information named as S1 table.

Reviewer #1: The authors propose a solution to a relevant clinical problem: intratumoral heterogeneity of glioma limits treatment choice and efficacy greatly, and radiomics indeed promises a relevant method to potentially tackle this problem as is stated by the authors and the references they refer to. MR-FLAIR can indeed be a very relevant sequence to use in this setting, especially with regards to invasiveness of the tumor. The study conducted is relevant for the field and quantitative MR imaging is currently very much studied within neuro-oncology. The manuscript could therefore be well suited for publication, however, there are some major remarks to be made at the manuscript in its current form:

1- IHC analysis

(1) The authors do not describe which antibodies are used for determination of protein expression, this should be mentioned for replication purposes. 

We feel great thanks for reviewer’s professional suggestion. 

The antibody information has been added in the “Immunohistochemical status” of the manuscript on Page 7. And the description was revised as follows:

 “We used rabbit anti-human monoclonal antibody for determination of Ki-67 (product number: RMA-0542) and vimentin (product number: RMA-0547) expression, used mouse anti-human monoclonal antibody for determination of CD34 (product number: Kit-0004) and S-100(product number: Kit-0007) expression. These preparations were all from Maixin Biotechnology Development Co., Ltd.”

 (2) It is stated that this was analyzed by two neuro-pathologists, however it is not explained what happens upon disagreement between these two pathologists.

The two pathologists independently reviewed all immunohistochemical results individually first and then reviewed together. Any discrepancies between two readers were discussed until a final consensus was generated. The information has been added in the “Immunohistochemical status” of the manuscript on Page 8.

2- Feature extraction: the ROIs that are used for feature extraction are not specified by the authors (tumor? edema? entire FLAIR abnormality?).

Thank you for this comment. The VOIs that are used for feature extraction are specified as entire FLAIR abnormality that drew along the edge of edema. Because FLAIR shows the whole tumor and edema clearly, if the tumor is enhanced, enhanced MRI can distinguish edema and tumor, but FLAIR can't distinguish them better.

We have also revised the description of the VOI delineation method in “Radiomics feature” part of “material and methods” on Page 8 as highlighted text: “The 3-dimensional VOIs (3D VOIs) were manually segmented by an experienced radiologist (9 years of radiology experience) using ITK-SNAP software (version 3.8.0, www.itksnap.org). The VOIs that were used for feature extraction were specified as entire FLAIR abnormality that drew along the edge of edema on 2-dimensional axial map of each layer and automatically merged into 3D VOIs.”

3- Statistical analysis: the authors only use their patient cohort to train radiomic feature models on. It would be recommended for the authors to look into validation methods for their models. It is understandable that external validation data would be difficult to come by but there are internal validation methods available which would provide more (reliable) data on the predictive power of these models. At this point, especially since the MRI sequences are obtained from one institute and one scanner, it could very well be that the models are only applicable for this specific institute. After internal validation this could of course still be the case but this would improve the quality of the results and interpretation. The results provided present the possibility of radiomics models to predict IHC expression, however some kind of validation would be highly recommended to strengthen the manuscript and the conclusions of the authors.

We really thank the reviewer’s professional comment for the model training, and we attempted to do some internal validation to strengthen the manuscript conclusion. 

For model validation, in addition to the 5-fold cross validation method as described in the method part of “Binary logistic regression models”, 3-fold cross validation and bootstrap methods, which are common approaches for model internal validation for small sample size were used and compared. The results are as follows.

 Table a1 The accurate score of Bootstrap validation method

Boot ROC 1st Qu Median Mean 3rd Qu

Ki67 0.456 0.534 0.530 0.615

CD34 0.561 0.667 0.646 0.736

S-100 0.823 0.871 0.863 0.913

Vimentin 0.757 0.815 0.807 0.875

 Table a2 The accurate score of 3-fold cross validation method

CV3 ROC 1st Qu Median Mean 3rd Qu

Ki67 0.440 0.493 0.486 0.551

CD34 0.594 0.667 0.639 0.698

S-100 0.752 0.757 0.800 0.823

Vimentin 0.792 0.813 0.813 0.833

 Table a3 The accurate score of 5-fold cross validation method.

CV5 ROC 1st Qu Median Mean 3rd Qu

Ki67 0.430 0.480 0.474 0.540

CD34 0.563 0.733 0.693 0.750

S-100 0.875 0.950 0.923 0.958

Vimentin 0.750 0.867 0.798 0.875

S2 Fig. ROC density for each model under bootstrap method

We have added texts in the method part of “Binary logistic regression models” on Page 11:

“In the current study, we only selected 51 patients with surgical and pathological diagnosis of gliomas, in which the immunohistochemistry of the four markers were not available for all. Therefore, we used and compared three kinds of validation methods including 3- and 5-fold cross validation and bootstrap, which are common approaches for model validation for small sample size [41,42].”

“For 3 or 5-folds repeat cross validation, all data were divided into 3 or 5 mutual exclusion subsets, 2 or 4 of which were used as training group data in turn and the remaining subset as validation data, then reselect different subsets as training and verification until all combinations of calendaring. Repeat above procedure 10 times. Additionally, bootstrap method repeat 100 times by sampling with return was also used. A total of 30, 50 or 100 accurate scores were obtained and the distribution (average, first quartile, third quartile) of score was used to evaluate the model performance so as to avoid model over-fitting.”

We have also added texts in the result part of “Evaluation of prediction models” on Page 14- Page 15:

“In the different internal validation method, the results of ROC show that the range of value transformation under different validation methods is almost the same. In addition, the relatively small deviation between median and mean value in each validation process reflected the logistic regression model is stable for the current dataset. For example, as shown in S2 Fig. in the supplementary information, the ROC for model of Ki-67, vimentin and S-100 under bootstrap method is dense at a special range. While the two separate peaks for the ROC density of CD34 model indicated the existing instability. Nevertheless, the validation results initially indicated that logistic regression model based on radiomics features could potentially predict the expression of the pathological biomarkers.”

Figure of “S2 Fig. ROC density for each model under bootstrap” method was added in the supplementary information. 

4- Patient cohort: did all patients included in this study receive a resection of their tumor? Often, glioma patients only receive a biopsy due to location of the tumor or physical condition of the patient. This could influence the IHC results obtained. Especially since the authors pose sampling-bias as a limitation of IHC analysis and radiomics as a solution to this it should be addressed what the IHC results are based upon.

Thank you for this valuable question and we feel sorry for making you confused about this. In the current study, the patients were selected who have received resection of the tumor for immunohistochemical assays. Therefore, the immunohistochemical analysis of the four biomarkers including Ki-67, vimentin, S-100 and CD34 were conducted based on the resected gliomas tumors.

We have revised the description in the method part of “Immunohistochemical status” on Page 7. 

Original text:

“We collected the glioma specimen surgically resect from patients’ brain for immunohistochemistry.”

Revised text:

“All the patients involved in the current study have received the operative resection of gliomas tumors. And the immunohistochemical analysis of the four biomarkers including Ki-67, vimentin, S-100 and CD34 were conducted based on the resected gliomas tumors.”

5- Language: This is a major concern for publication. The manuscript contains a lot of grammatical and spelling errors, this makes parts of the manuscript hard to read and fully comprehend. Without optimization of the use of language this manuscript is not suitable for publication. The authors have an incoherent use of abbreviations (some are spelled out multiple times, abbreviation is sometimes used and sometimes not and different abbreviations are used i.e. VOI/ROI). It is recommended to review this manuscript on the correct use of English, potentially by an external partner or language center before publication.

We feel very sorry for the errors in the language management! We have double checked and corrected the errors in the whole manuscript, including the grammar, spelling and confusing or incoherent abbreviations. The revised parts were recorded in the tracked-version of the manuscript.

6- Clinical relevance: 

(1) The authors state that they want to identify immunohistochemical subtypes to aid precision medicine. However, the manuscript insufficiently states the clinical consequences of identifying these specific markers non-invasively. How can identification of these markers aid in the treatment plan that is clinically applied?

We appreciated the reviewer’s careful consideration and learned a lot during reorganizing our idea. The tumor characters of heterogeneity and multimolecular synergy promotes the development of non-invasive imaging tools for tumor-associated molecules identification or tracing. In the current study, we try to use MRI-derived radiomics features to initially explore if the radiomics-assisted tools could help identifying or predicting four gliomas-related molecular markers, including Ki-67, vimentin, S-100 and CD34, to rule out their existence or not, high or low expression. For the aspect of immunohistochemical diagnosis, except for Ki-67, which has been included in WHO of CNS tumor routinely. The other three markers of vimentin, S-100 and CD34 have been studied rarely for gliomas as a single specific marker and frequently utilized as co-staining histological markers for differential diagnosis or prognostic prediction. For example, the combination of GFAP, EMA, S100 and vimentin was used to assist epithelioid glioblastoma (Ep-GBM) which is one provisional new variant of glioblastoma added to the WHO 2016 classification [9]. Vimentin is often co-stained with GFAP, Ki-67 and p53 for diffuse astrocytoma as its enhancement factors for cell motility and invasion [17]. S-100 is quite useful in the diagnosis of poorly differentiated tumors. So S-100 is often involved in most of glioblastomas immunochemical diagnosis [10]. CD34 is popular as a vessel marker and is demonstrated to regulate the glioma angiogenesis and could help gliomas grading [19]. These diagnosis guidelines or studies indicate that fully consider the molecular pathology may greatly help confirm gliomas sub-type and prognostic prediction. Therefore, we would like to take the first step to validate if the simple imaging-based radiomics method could help molecular pathology subtyping and assist the routine histological assays when multiple biomarkers need to be evaluated for glioma patients.

The clinical relevance for the respective marker was described in detail as follows.

Ki-67: The nuclear protein Ki-67 has been widely used as an indicator of cell proliferation in gliomas. Several meta-analysis indicated that high level of Ki-67 expression was an important predictive factor for poor prognosis of glioma patients [16].

Vimentin: Vimentin is a primary intermediate filament protein, which is functional in protein organization involved in adhesion, migration and cell signaling. Upregulation of vimentin has been increasingly involved in Epithelial-mesenchymal transition (EMT) [Thiery JP. Epithelial-mesenchymal transitions in tumour progression. Nature reviews. Cancer. 2002, 2:442-454.], a process closely associates with development, wound healing and cancer metastasis [Battaglia RA, Delic S, Herrmann H, Snider NT. Vimentin on the move: new developments in cell migration. F1000Res. 2018,7. [PMID: 30505430 DOI: 10.12688/f1000research.15967.1]. 

The revised discussion part on Page 27-Page 28 was as follows: “The overexpression of vimentin has been demonstrated to be associated with an increased migratory or invasive capacity of the cancer cells [59]. Therefore, vimentin is considered as a potential indicator for cancer prognosis and therapeutic target [60]. Vimentin was postulated as a molecular marker presenting enhanced motility and invasion in gliomas [17,18,61]. Its expression indicates a lower degree of differentiation and was found in high-grade astrocytomas [46]. As vimentin indicated enhanced invasion and cells of invasive gliomas show a decreased proliferation rate and a relative resistance to apoptosis [18], high vimentin expression could be taken as a prognostic factor for treatment difficulty or poor survival in diffusive glioma or high-grade glioma patients. Lin et al [12] systemically analyzed the vimentin expression and found that vimentin expression was associated with tumor grade and overall survival of high-grade glioma patients. High vimentin expression was an independent significant prognostic factor for poor survival in high-grade glioma patients while low vimentin expression a biological indicator of better response to temozolomide therapy for glioblastoma patients [19]. It is encouraging that T2 FLAIR radiomics could be further explored to validate its usefulness in imaging-based assessment of migratory, invasion and related prognostication for different gliomas subtypes.”

S-100: The S-100 proteins belongs to calcium-binding protein family of EF-hand type (helix E-loop-helix F), which is composed of 25 known members [Xia C, Braunstein Z, Toomey AC, Zhong J, Rao X. S100 Proteins As an Important Regulator of Macrophage Inflammation. Front Immunol. 2017,8:1908.]. S-100 proteins are involved in tumor occurrence and development including regulation of cell differentiation, cell cycle progression, cell proliferation, cell apoptosis, cell motility, invasion and migration, tumor microenvironment and cancer stem cells (CSCs) [54]. Several studies revealed that S-100 protein members were related to gliomas subtype identification, tumor progression and therapy effectiveness. Camby et al [55] indicated that several S-100 proteins could help differential diagnosis or judge malignancy of human astrocytic tumors. For example, modifications in the level of S100A3 protein expression level could help identify the pilocytic astrocytomas from WHO grade II-IV astrocytic tumors. While S100A6 protein expression enabled a clear distinction between low (WHO grade I and II) and high (WHO grade III and IV) grade astrocytic tumors. S100A1, S100A4 and S100B protein expression are related to the increasing levels of tumor malignancy. Another study reported that S100A4 played a crucial role in neutrophil-promoting tumor progression and S100A4 depletion could increase the effectiveness of anti-VEGF therapy in glioma [56].” For the prognosis value of S-100 proteins, there is also a study reported that transcript levels of S100A8/S100A9 could be independent poor prognostic indicators for GBM [58]. Serum S100 protein levels were also studied as prognostic markers for glioma patients. For example, serum S100B level might be valuable in prognostication of survival for recurrent glioma patients [57], while medium levels of pre-operative and three-month post-operative serum S100A8 levels could help predict poor prognosis in GBM patients. 

CD34: CD34 is a transmembrane phosphoglycoprotein and well known as a biomarker for vascular density and angiogenesis. The use of CD34 for the prognosis, diagnosis, and treatment of various cancers has been increasing. The precise identification of CD34 noninvasively could help predicting angiogenesis-related gliomas progression [5-8]. Beside grading, CD34 expression might of assistance to indicate glioblastoma stem-like cells differentiation into tumor-associated endothelial cells in low-grade gliomas [62]. In addition, CD34 might have strong potential in predicting gliomas survival and therapy effectiveness [20]. CD34 staining were also associated with hypoxia-induced angiogenesis and may play a role in glioblastoma hemorrhage. Its underlying mechanism of which may promote the development individualized therapies for glioblastoma [63].

In order to make the objective of the current study clearer to the readers, we have revised the description for the reason for choosing these four biomarkers as research objects in the introduction part from Page 3- Page 4.

(2) It would be very helpful if the authors could provide data on the prognostic relevance of these IHC markers in their patient cohort to see whether identification of these markers can help in predicting patient prognosis.

Thank you very much for this valuable and practical suggestion.

Based on the small amount of data collected, this paper mainly focused on if the radiomics-assisted tools could help identifying or predicting four gliomas-related molecular markers. Because this study only contains a small amount of follow-up information, it is not suitable for prognosis or survival analysis. In the next step, we will expand the sample size and include more follow-up information to conduct research on prognosis prediction of four biomarkers based on the radiomics.

(3) In the discussion the authors correctly refer to the new WHO classification of CNS tumors which incorporate molecular markers but do not clarify what the relevance of this is compared to their work and vice versa.

Thank you very much for this kind reminder. We deleted the content about gene phenotype, because in our study radiomics has been used to predict for proteomic (e.g., Ki-67, CD34, S-100, vimentin expression), and the relationship between radiomics and genomics (such as IDH1 status) was not involved. And we have made some revision for this discussion part from two aspects including: (a)Why we want to have a try to use radiomics for identifying the four biomarkers of Ki-67, vimentin, S-100 and CD34. (b)How about the diagnosis effectiveness of such radiomics-based method.

(a) Why we want to have a try to use radiomics for identifying the four biomarkers of Ki-67, vimentin, S-100 and CD34?

The WHO 2016 classification of CNS tumors introduces both of the phenotype and the genotype into daily diagnosis and makes it possible to lead a more precise diagnosis of the various entities toward a personalized management of brain tumors. Besides the genomic markers mentioned in the WHO 2016, the concept of molecular diagnosis also promotes immunohistochemical assays to involve more molecular biomarkers besides those for traditional gliomas grading, leading to more precise pathological subtype and better prognostic prediction [9,10]. For example, in our study, the second and third cases in Fig 8 were all glioma grade II. Although pathological confirmed same grade, their immunohistochemical expressions were extremely different. In the second case, the expression of S-100 and CD34 was negative, but in the third case, the expression of S-100 and CD34 was positive; the expression of vimentin was positive in the second case while the negative in the third case. Ki-67 expression was positive in both cases. Therefore, fully considering the molecular pathology may greatly help confirming gliomas heterogenous sub-type and prognostic prediction. Exploring more biomarkers related to gliomas molecular typing, and the association of imaging-based radiomics features with these biomarkers is conducive to realize easy-to-implement radiomics-assisted multi-biomarker “histological” analysis and promote the personalized treatment of gliomas. However, the first step is to validate if the radiomics is effective in identifying the molecular biomarkers. So in the current study, we try to use radiomics features to initially identify four markers of Ki-67, vimentin, S-100 and CD34, which are histological markers recommended in several gliomas diagnosis [9,10,20,46]. 

b) How about the diagnosis effectiveness of such radiomics-based method?

In summary, we have established four immunohistochemical predicting models for four kinds of immunohistochemical biomarkers (Ki-67, S-100, vimentin and CD34) in gliomas patients based on T2 FLAIR radiomics, which are potential for imaging-based prediction of tumor proliferation, malignancy, therapeutic effectiveness, migratory or invasion, and angiogenesis. Ki-67, S-100 and vimentin models have higher reliability and can effectively predict the expression of respective protein.

“The best prediction performance was observed in the S-100 model (AUC: 0.920; sensitivity: 89.29%; specificity: 90.48%; accuracy: 82.00%). The good S-100 prediction performance of T2 FLAIR radiomics in the current study implies its potential in the future applications in gliomas treatment decision-making and effectiveness assessment.”

“The vimentin model (AUC: 0.854; sensitivity: 87.50%; specificity: 72.22%; accuracy: 73.80%) had good prediction performance as well. It is encouraging that T2 FLAIR radiomics could be further explored to validate its usefulness in imaging-based assessment of migratory, invasion and related prognostication for different gliomas subtypes.”

“The Ki-67 model exhibited moderate performance (AUC: 0.713; sensitivity: 69.23%; specificity: 66.67%; accuracy: 66.00%). In our results, although the comprehensive Ki-67 prediction efficiency of T2 FLAIR radiomics is moderate, it might introduce additional power for prediction of tumor proliferative behavior when combined with other conventional or functional MRI series.”

“The CD34 model (AUC, 0.745; sensitivity, 55.56%; specificity, 87.50%; accuracy, 82.00%) had lower sensitivity while better specificity and accuracy. Therefore, our result indicates that T2 FLAIR radiomics might provide a noninvasive identifier of angiogenesis, which is useful for gliomas progression or prognosis analysis. However, the low sensitivity of the current model may need further consideration and involving larger samples.”

“According to the standard net benefit of the decision-making curve, it could be found that within a relatively large threshold range, the Ki-67, S-100 and vimentin models were applicable for decision-making. The CD34 model had weak reference value except for threshold probability between 0.75-0.9. Therefore, Ki-67, S-100 and vimentin models were proposed for immunohistochemical biomarker prediction. While the CD34 model was not weaker for positive and negative discrimination, which may be related to the data distribution included in this study.”

We have revised the discussion and added the clinical relevance of four biomarkers in the manuscript according to the reviewer’s comments and suggestions, as from Page 27 to Page 30.

Reviewer #2: General: This work proposes to use T2 FLAIR radiomic features in predicting the immunohistochemical subtype of glioma like Ki-67, S-100, vimentin and CD34. This work uses only limited dataset of 51 glioma patient samples which is not enough to validate the findings. Here some comments:

1-Abstract: Author mentioned the “The deep radiomics” this is mean that the authors used deep learning which is not the case in this paper. I suggest to use only the radiomics.

Many thanks to the reviewer’s professional and kind suggestion. We have substitute “The deep radiomics” into “The radiomics”.

2-Introduction:

(1)-The authors did a good state of the art regarding the immunohistochemical subtype. However, we suggest the authors to involve a section about radiomics for brain tumor and involve the following citations:

1-Integration of Radiomic and Multi-omic Analyses Predicts Survival of Newly Diagnosed IDH1 Wild-Type Glioblastoma

2-Predicting the gene status and survival outcome of lower grade glioma patients with multimodal MRI features

3-Radiomics in glioblastoma: current status and challenges facing clinical implementation

Thank you very much for your careful review, we really learn a lot. After carefully learning the mentioned articles, we have involved these citations in the introduction part on page 

The revised texts are as follows on Page 4.

“Abundant information extracted from radiomics features give us chances to establish bridge between radiological image features and tumor-associated molecules [22]. It is appearing more and more radiomics-based tools for studying gliomas. Radiomics features derived from MRI images have been used to predict grades of gliomas and showed good performance [23,24]. In addition, texture analysis based on MR images has also been applied in molecular or genomic subtyping and their survival outcome relevance for gliomas [25-27]. The previous studies for molecular or gene subtyping utilized single- or multi-modal MRI features extracted from sequences of T1WI, T2WI, T2 FLAIR, Contrast-enhanced T1-weighted images (T1-CE), advanced MRI techniques such as diffusion-weighted imaging (DWI) and arterial spin labeling (ASL).”

(2) In addition, showing the differences with these literatures and how the proposed work is different.

Thank you very much for the reviewer’s valuable suggestion and we reorganized our idea and revised the introduction part from Page 3 to Page 5 according to the two points to illustrate how we proposed the current work as follows.

(a) Why do we use T2 FLAIR images to extract radiomics features?

T2 FLAIR is one of principle imaging sequence for assessment of gliomas and has the best contrast between infiltrating tumor margins and normal brain [28,29]. And the texture analysis based on T2 FLAIR images has been demonstrated to be helpful and accurate in predicting IDH mutation [30,31]. T2 FLAIR features also showed distinct characteristics between IDH wildtype and mutant tumors,1p/19q co-deleted and 1p/19q intact tumors, MGMT methylated and unmethylated tumors respectively, which could useful in molecular classification of patients with grade II/III glioma [32]. In addition, T2 FLAIR is increasingly involved in the texture analysis for assessment of therapeutic response and survival outcomes. It has been reported that 3D shape features could distinguish pseudoprogression from true progression and lower edge contrast of the FLAIR signal is correlated with poor survival after bevacizumab treatment [33,34].

All of these indicate that T2 FLAIR is candidate to capture some features in the non-enhanced state, and the visible tumor phenotypic or genotypic characteristics can be systematically quantified [35]. Therefore, radiomics analysis of pre-operative T2 FLAIR data is principally functional for identification of immunohistochemical biomarkers, which is often crucial for gliomas pathological sub-typing and prognosis predicting. However, the T2 FLAIR-based texture analysis was rarely studied in glioma immunohistochemical typing research. At present, considerable Ki-67 radiomics studies can be detected [36]. While S-100, CD34 and vimentin have not been discussed yet. In this study, we proposed radiomics features and the binary logistic regression model to identify the immunohistochemical typing of Ki-67, S-100, CD34 and vimentin, so as to achieve the image-indication of tumor progression, angiogenesis, proliferation or invasion. We hypothesized that T2 FLAIR-based radiomics methods would facilitate imaging subtypes of gliomas that relate to prognosis and underlying molecular characteristics of the tumor.

(b)Why we want to use radiomics features to identify the four biomarkers of Ki-67, vimentin, S-100 and CD34?

 The WHO 2016 classification of CNS tumors also introduces both of the phenotype and the genotype into daily diagnosis and makes it possible to lead a more precise diagnosis of the various entities toward a personalized management of brain tumors. Besides the genomic markers mentioned in the WHO 2016, the concept of molecular diagnosis also promotes immunohistochemical assays to involve more molecular biomarkers besides those for traditional gliomas grading, leading to more precise pathological subtype and better prognostic prediction [9,10].

In the current study, we try to use MRI-derived radiomics features to initially explore if the radiomics-assisted tools could help identifying or predicting four gliomas-related molecular markers, including Ki-67, vimentin, S-100 and CD34, to rule out their existence or not, high-or low expression. For the aspect of immunohistological diagnosis, except for Ki-67, which has been included in WHO of CNS tumor routinely, the other three markers of vimentin, S-100 and CD34 have been studied rarely for gliomas as a single specific marker and frequently utilized as co-staining histological markers for differential diagnosis or prognostic prediction. For example, the combination of GFAP, EMA, S100 and vimentin was used to assist epithelioid glioblastoma (Ep-GBM) which is one provisional new variant of glioblastoma added to the WHO2016 classification [9]. Vimentin is often co-stained with GFAP, Ki-67 and p53 for diffuse astrocytoma as its enhancement factors for cell motility and invasion [17]. While vimentin is also considered in diagnosis of giant cell glioblastoma. S100 is quite useful in the diagnosis of poorly differentiated tumors. So S100 is often involved in most of glioblastomas immunohistological diagnosis [10]. CD34 is popular as a vessel marker indicating the microvascular density of various tumors [19]. These diagnosis guidelines or studies indicate that fully consider the molecular pathology may greatly help confirm gliomas sub-type and prognostic prediction. Therefore, we would like to take the first step to validate if the simple imaging-based radiomics method could help molecular pathology subtyping and assist the routine histological assays when multiple biomarkers need to be evaluated for glioma patients.

The clinical relevance for the respective marker was described in detail as follows:

Ki-67: The nuclear protein Ki-67 has been widely used as an indicator of cell proliferation in gliomas. Several meta-analysis indicated that high level of Ki-67 expression was an important predictive factor for poor prognosis of glioma patients [16].

Vimentin: Vimentin is a primary intermediate filament protein, which is functional in protein organization involved in adhesion, migration and cell signaling. Upregulation of vimentin has been increasingly involved in Epithelial-mesenchymal transition (EMT) [Thiery JP. Epithelial-mesenchymal transitions in tumour progression. Nature reviews. Cancer. 2002, 2:442-454.], a process closely associates with development, wound healing and cancer metastasis [Battaglia RA, Delic S, Herrmann H, Snider NT. Vimentin on the move: new developments in cell migration. F1000Res. 2018,7. [PMID: 30505430 DOI: 10.12688/f1000research.15967.1]. 

The revised discussion part on Page 27-Page 28 was as follows: “The overexpression of vimentin has been demonstrated to be associated with an increased migratory or invasive capacity of the cancer cells [59]. Therefore, vimentin is considered as a potential indicator for cancer prognosis and therapeutic target [60]. Vimentin was postulated as a molecular marker presenting enhanced motility and invasion in gliomas [17,18, 61]. Its expression indicates a lower degree of differentiation and was found in high-grade astrocytomas [46]. As vimentin indicated enhanced invasion and cells of invasive gliomas show a decreased proliferation rate and a relative resistance to apoptosis [18], high vimentin expression could be taken as a prognostic factor for treatment difficulty or poor survival in diffusive glioma or high-grade glioma patients. Lin et al [12] systemically analyzed the vimentin expression and found that vimentin expression was associated with tumor grade and overall survival of high-grade glioma patients. High vimentin expression was an independent significant prognostic factor for poor survival in high-grade glioma patients while low vimentin expression a biological indicator of better response to temozolomide therapy for glioblastoma patients [19]. It is encouraging that T2 FLAIR radiomics could be further explored to validate its usefulness in imaging-based assessment of migratory, invasion and related prognostication for different gliomas subtypes.”

S-100: The S-100 proteins belongs to calcium-binding protein family of EF-hand type (helix E-loop-helix F), which is composed of 25 known members [Xia C, Braunstein Z, Toomey AC, Zhong J, Rao X. S100 Proteins As an Important Regulator of Macrophage Inflammation. Front Immunol. 2017,8:1908.]. S-100 proteins are involved in tumor occurrence and development including regulation of cell differentiation, cell cycle progression, cell proliferation, cell apoptosis, cell motility, invasion and migration, tumor microenvironment and cancer stem cells (CSCs) [54]. Several studies revealed that S-100 protein members were related to gliomas subtype identification, tumor progression and therapy effectiveness. Camby et al [55] indicated that several S-100 proteins could help differential diagnosis or judge malignancy of human astrocytic tumors. For example, modifications in the level of S100A3 protein expression level could help identify the pilocytic astrocytomas from WHO grade II-IV astrocytic tumors. While S100A6 protein expression enabled a clear distinction between low (WHO grade I and II) and high (WHO grade III and IV) grade astrocytic tumors. S100A1, S100A4 and S100B protein expression are related to the increasing levels of tumor malignancy. Another study reported that S100A4 played a crucial role in neutrophil-promoting tumor progression and S100A4 depletion could increase the effectiveness of anti-VEGF therapy in glioma [56]. For the prognosis value of S100 proteins, there is also a study reported that transcript levels of S100A8/S100A9 could be independent poor prognostic indicators for GBM [58]. Serum S100 protein levels were also studied as prognostic markers for glioma patients. For example, serum S100B level might be valuable in prognostication of survival for recurrent glioma patients [57], while medium levels of pre-operative and three-month post-operative serum S100A8 levels could help predict poor prognosis in GBM patients. 

CD34: CD34 is a transmembrane phosphoglycoprotein and well known as a biomarker for vascular density and angiogenesis. The use of CD34 for the prognosis, diagnosis, and treatment of various cancers has been increasing. The precise identification of CD34 noninvasively could help predicting angiogenesis-related gliomas progression [5-8]. Beside grading, CD34 expression might of assistance to indicate glioblastoma stem-like cells differentiation into tumor-associated endothelial cells in low-grade gliomas [62]. In addition, CD34 might have strong potential in predicting gliomas survival and therapy effectiveness [20]. CD34 staining were also associated with hypoxia-induced angiogenesis and may play a role in glioblastoma hemorrhage. Its underlying mechanism of which may promote the development individualized therapies for glioblastoma [63].

In order to make the objective of the current study clearer to the readers, we have revised the introduction part on Page 3 and Page 4.

In addition, we also revised the discussion part on page 27-30 to clearly describe the effectiveness of the radiomics methods in our work for identifying such four biomarkers and the respective clinical relevance.

3-Materials and methods:

(1)- how the authors apply the normalization? We suggest authors to clarify all the preprocessing steps (normalization, resolution, field corrections…etc.).

Thank you very much for your question. Image preprocessing is the part that we missed in the article. We have added texts in the method part of “Radiomics feature” (on Page 8) as follows: 

“Before feature extraction, z-score standardization was applied to images. Since the data is single center and the scanning consistency is good, resampling and bias field correction are not adopted in this study.”

(2)-please clarify the number of samples for each of the labels to estimate the % for class 1 and class 2

Thanks for your advice, the number of samples for each labels of different class were listed in the Table 2.

(3)-How the authors compute these radiomic functions more detail is recommended.

Thanks for your advice, we extract radiomics features using AK software, the mathematic functions content of radiomics features were add into the supplementary material.

(4)-Why the author chooses the logistic regression for classification and no other classifier model like random forest, SVM, CNN?

We really thank the reviewer’s professional comment for the model training.

The rest of the algorithm is used in large sample sizes and requires validation set. However, in the current study, we only selected 51 patients with surgical and pathological diagnosis of gliomas, in which the immunohistochemistry of the four markers were not available for all. Therefore, the logistic regression method was chosen. For model validation, in addition to the 5-fold cross validation method as described in the method part of “Binary logistic regression models”, 3-fold cross validation and bootstrap methods, which are common approaches for model internal validation for small sample size were used and compared. The results are described as follows:

 Table a1 The accurate score of Bootstrap validation method

Boot ROC 1st Qu Median Mean 3rd Qu

Ki67 0.456 0.534 0.530 0.615

CD34 0.561 0.667 0.646 0.736

S-100 0.823 0.871 0.863 0.913

Vimentin 0.757 0.815 0.807 0.875

 Table a2 The accurate score of 3-fold cross validation method

CV3 ROC 1st Qu Median Mean 3rd Qu

Ki67 0.440 0.493 0.486 0.551

CD34 0.594 0.667 0.639 0.698

S-100 0.752 0.757 0.800 0.823

Vimentin 0.792 0.813 0.813 0.833

 Table a3 The accurate score of 5-fold cross validation method.

CV5 ROC 1st Qu Median Mean 3rd Qu

Ki67 0.430 0.480 0.474 0.540

CD34 0.563 0.733 0.693 0.750

S-100 0.875 0.950 0.923 0.958

Vimentin 0.750 0.867 0.798 0.875

Fig. S2 ROC density for each model under bootstrap method

We have added texts in the method part of “Binary logistic regression models” on Page 11:

“In the current study, we only selected 51 patients with surgical and pathological diagnosis of gliomas, in which the immunohistochemistry of the four markers were not available for all. Therefore, we used and compared three kinds of validation methods including 3- and 5-fold cross validation and bootstrap, which are common approaches for model validation for small sample size [41,42].”

“For 3 or 5-folds repeat cross validation, all data were divided into 3 or 5 mutual exclusion subsets, 2 or 4 of which were used as training group data in turn and the remaining subset as validation data, then reselect different subsets as training and verification until all combinations of calendaring. Repeat above procedure 10 times. Additionally, bootstrap method repeat 100 times by sampling with return was also used. A total of 30, 50 or 100 accurate scores were obtained and the distribution (average, first quartile, third quartile) of score was used to evaluate the model performance so as to avoid model over-fitting.”

We have also added texts in the result part of “Evaluation of prediction models” on Page 14:

“In the different internal validation method, the results of ROC show that the range of value transformation under different validation methods is almost the same. In addition, the relatively small deviation between median and mean value in each validation process reflected the logistic regression model is stable for the current dataset. For example, as shown in S2 Fig. in the supplementary information, the ROC for model of Ki-67, vimentin and S-100 under bootstrap method is dense at a special range. While the two separate peaks for the ROC density of CD34 model indicated the existing instability. Nevertheless, the validation results initially indicated that logistic regression model based on radiomics features could potentially predict the expression of the pathological biomarkers.”

Figure of “S2 Fig. ROC density for each model under bootstrap method” was added in the supplementary information. 

 (5)-I suggest the authors to apply multivariate analysis by dividing the patient’s data to 75 % training and 25 % testing and measure the performance metrics to avoid the doubt about over fitting.

We thank the reviewer’s professional advice. As the reply for the last comments, different internal validation methods were adopted for our small sample size in the current study. We would divide the data into training and testing set according to the reviewer’s suggestion in our future study when more data are collected.

4-In discussion: I suggest authors to avoid the first proposed radiomics in “Since the Dutch scholar Lambin et al [36] first proposed radiomics in 2012，high throughput textures analysis as a new technology has been applied to the studies of tumors diagnosis, staging, grading and predictive treatment more extensively.” Because there are many papers published before 2012 using the term of features extraction for predicting the cancer grade.etc.

 Thank you very much for your rigorous and kind suggestion. We have revised the description by deleting the word of “first” according to the reviewer’s comments.

---

## [Editor Report · Decision Letter 1]

27 Dec 2019

High-order radiomics features based on T2 FLAIR MRI predict multiple glioma immunohistochemical features: a more precise and personalized gliomas management

PONE-D-19-22703R1

Dear Dr. Liu,

We are pleased to inform you that your manuscript has been judged scientifically suitable for publication and will be formally accepted for publication once it complies with all outstanding technical requirements.

With kind regards,

Alessandro Weisz

Academic Editor

PLOS ONE

Additional Editor Comments (optional):

The A.s thoroughly revised the manuscript taking into account of all comments and suggestions raised by the two R.s. As a result, the main weaknesses and unclear aspects of the study, identified during review, were corrected and the overall quality of the manuscript results significantly improved. For this reason the revised manuscript is in my opinion acceptable for publication.
---

## [Editor Report · Acceptance letter]

9 Jan 2020

PONE-D-19-22703R1 

High-order radiomics features based on T2 FLAIR MRI predict multiple glioma immunohistochemical features: a more precise and personalized gliomas management 

Dear Dr. Liu:

I am pleased to inform you that your manuscript has been deemed suitable for publication in PLOS ONE. Congratulations! Your manuscript is now with our production department. 

With kind regards,

on behalf of

Dr. Alessandro Weisz 

Academic Editor

PLOS ONE